# MagicTryOn: Harnessing Diffusion Transformer for Garment-Preserving Video Virtual Try-on

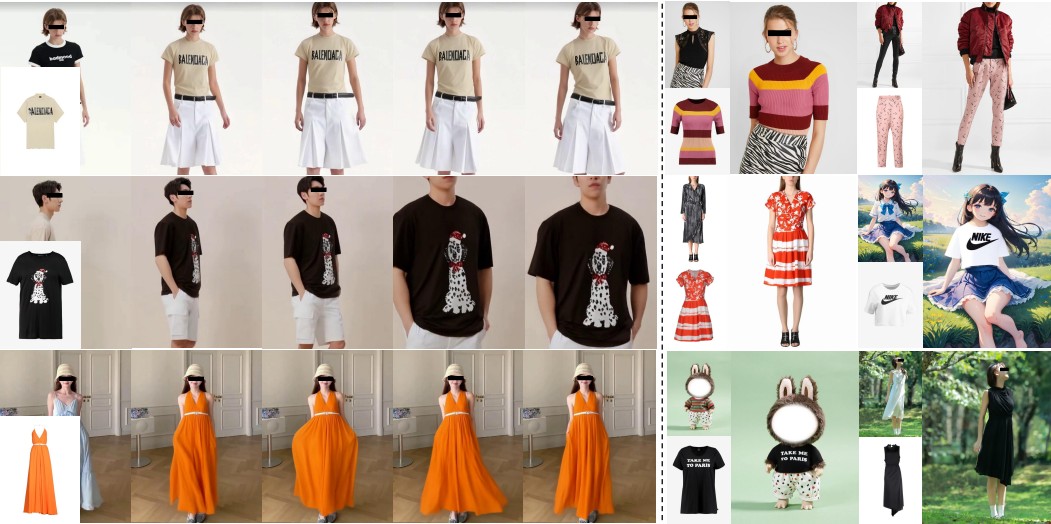

Figure 1: MagicTryOn can accurately transfer the target garment onto the target person under unconstrained settings, while preserving spatiotemporal consistency and high garment fidelity throughout multi-pose video sequences.

## Abstract

Video Virtual Try-On (VVT) aims to synthesize garments that appear natural across consecutive video frames, capturing both their dynamics and interactions with human motion. Despite recent progress, existing VVT methods still suffer from inadequate garment fidelity and limited spatiotemporal consistency. The reasons are (i) under-exploitation of garment information, with limited garment cues being injected, resulting in weaker fine-detail fidelity, and (ii) the lack of spatiotemporal modeling, which hampers cross-frame identity consistency and causes temporal jitter and appearance drift. In this paper, we present Magic-TryOn, a diffusion transformer–based framework for garment-preserving video virtual try-on. To preserve fine-grained garment details, we propose a fine-grained garment-preservation strategy that disentangles garment cues and injects these decomposed priors into the denoising process. To improve temporal garment consistency and suppress jitter, we introduce a garment-aware spatiotemporal rotary positional embedding (RoPE) that extends RoPE within full self-attention, using spatiotemporal relative positions to modulate garment tokens. We further impose a mask-aware loss during training to enhance fidelity within garment regions. Moreover, we adopt distribution-matching distillation to compress the sampling trajectory to four steps, enabling real-time inference without degrading garment fidelity. Extensive quantitative and qualitative experiments demonstrate that MagicTryOn outperforms existing methods, delivering superior garment-detail fidelity and temporal stability in unconstrained settings. Code will be made publicly available.

# 1 INTRODUCTION

Video Virtual Try-On (VVT) aims to simulate the realistic appearance of individuals wearing garments across consecutive video frames, capturing the natural look of garments in dynamic environments and their complex interactions with human movements. Compared to image-based virtual try-on (Choi et al., 2024; Gou et al., 2023; Kim et al., 2024; Morelli et al., 2023; Xu et al., 2025; Wan et al., 2024; Jiang et al., 2024; Li et al., 2025b), video virtual try-on exhibits greater capabilities and application potential in presenting garment motion variations and deformation responses.

Recently, several methods (Fang et al., 2024; He et al., 2024; Xu et al., 2024a; Wang et al., 2024b; Li et al., 2025c; Nguyen et al., 2025; Chong et al., 2025; Zuo et al., 2025) have been proposed specifically for the VVT task. These methods typically build on pretrained diffusion models and inject garment information into the denoising network. For example, they employ stable video diffusion (Blattmann et al., 2023a) or multi-modal diffusion transformer (Li et al., 2024) as the backbone, inject garment information via a reference network or low-rank adaptation modules, and then fuse it into the main sequence through feature concatenation. Although they achieve notable results, they still show limitations in garment fidelity and spatiotemporal consistency. The reasons are two-fold. (i) They under-exploit garment information, and the injected cues are limited, which constrains the network's ability to preserve fine details. In practice, a single garment image or a text caption is often injected via feature concatenation, without explicitly leveraging complementary cues, resulting in limited use of garment information. Intuitively, decomposing the garment image into complementary semantic, structural, and appearance cues and injecting them jointly into the denoising network can improve garment preservation. (ii) They lack spatiotemporal modeling of garment features and garment-specific positional encoding, which prevents self-attention from consistently aligning the same garment across frames. As a result, the network struggles to maintain a stable garment identity under deformation, leading to temporal jitter and appearance drift.

Based on the above analysis, we propose MagicTryOn, a diffusion transformer–based framework for garment-preserving video virtual try-on. To tackle the under-exploitation of garment information, we introduce a fine-grained garment-preservation strategy. Specifically, we decompose garment cues into three complementary streams (semantics, structure, and appearance) and inject them into the denoising network. The semantics stream encodes the garment's category, attributes, material, and color. The structure stream encodes the garment's silhouette and topology. The appearance stream encodes the garment's detail features. To address temporal jitter caused by the lack of spatiotemporal modeling, we improve the rotary position embedding (RoPE) within full self-attention by extending it to a garment-aware spatiotemporal RoPE. We apply spatiotemporal relative position modulation to garment token, explicitly characterizing the relative relations and correspondence constraints of the same garment under cross-frame deformation. In addition, to further enhance the model's ability to preserve garments, we introduce a mask-aware loss during training to strengthen the optimization of garment regions. Furthermore, to meet the demands of scenarios requiring faster inference, we apply distribution matching distillation to MagicTryOn, reducing the inference steps to four and accelerating the inference speed by $50\times$ while maintaining try-on quality. Our contributions to the community are threefold:

(i) We propose MagicTryOn and improve garment preservation by decomposing garment cues into semantics, structure, and appearance, injecting them into the denoising network, and introducing a mask-aware loss that focuses optimization on garment regions.

(ii) We extend RoPE to a garment-aware spatiotemporal RoPE, providing explicit cross-frame correspondence constraints that reduce temporal jitter and preserve garment identity under deformation.

(iii) We apply distribution-matching distillation to compress inference to four steps, achieving 50× speed-up while maintaining try-on quality. Extensive experiments show that our MagicTryOn surpasses state-of-the-art approaches on standard metrics and visual quality.

# 2 RELATED WORK

## 2.1 VIDEO VIRTUAL TRY-ON

Compared to image virtual try-on (Kim et al., 2024; Wan et al., 2024; Jiang et al., 2024; Liang et al., 2024; Li et al., 2025b; Xu et al., 2025; Luo et al., 2025; Zhou et al., 2025; Luan et al., 2025), video

virtual try-on (VVT) enables more natural and fluid try-on experiences for users. Current methods (Fang et al., 2024; He et al., 2024; Xu et al., 2024a; Li et al., 2025c; Nguyen et al., 2025; Wang et al., 2024b; Karras et al., 2024; Zuo et al., 2025) predominantly leverage diffusion models for VVT tasks. For instance, WildVidFit (He et al., 2024) generated video try-on results using image-guided controllable diffusion models, replacing explicit warping operations with a detail-oriented single-stage image try-on network to alleviate occlusion issues. ViViD (Fang et al., 2024) adapted image diffusion models to video tasks by introducing temporal modeling modules and designed a garment encoder to extract fine-grained semantic features of clothing. RealVVT (Li et al., 2025c) proposed a photorealistic video virtual try-on framework to enhance stability and realism in dynamic video scenes. CatV$^2$TON (Chong et al., 2025) adopted a video DiT architecture to unify image and video try-on within a single diffusion model. DPIPM (Li et al., 2025a) leveraged diffusion modeling to explicitly capture dynamic pose interactions, advancing video virtual try-on. However, they remain limited under complex cases such as multi-garment scenarios, as they fail to fully exploit garment information and lack spatiotemporal modeling for garments. To overcome these challenges, we design MagicTryOn to enhance generation performance in complex cases.

## 2.2 Video Generation

Video generation methods based on diffusion models can be broadly categorized into two groups, Text-to-Video (T2V) (Blattmann et al., 2023b; Deng et al., 2023; Guo et al., 2023; Yang et al., 2024; Menapace et al., 2024; Ren et al., 2024; Jeong et al., 2024) and Image-to-Video (I2V) (Hu, 2024; Xu et al., 2024b; Zeng et al., 2024; Guo et al., 2024; Shi et al., 2024; Zhang et al., 2024; Niu et al., 2024). For instance, AnimateDiff (Guo et al., 2023) introduced a plug-and-play motion modeling module that seamlessly integrates with personalized text-to-image models to enable animation generation. Tune-A-Video (Wu et al., 2023) enhanced temporal consistency by strengthening self-attention mechanisms to jointly reference previous frames and the initial frame during current frame synthesis. VideoPainter (Bian et al., 2025) incorporated a lightweight contextual encoder to generalize across various types of occlusions. As a specialized form of video generation, video virtual try-on requires synthesizing given garments onto appropriate regions of dynamically moving humans, while simultaneously preserving garment details and styles and ensuring spatial and temporal consistency in the generated videos. To address these unique challenges, in this paper, we specifically design a DiT-based generative framework tailored for video virtual try-on.

## 3 Methodology

Our method aims to tame the pretrained diffusion transformer (DiT) for video virtual try-on, addressing the common issues of temporal jitter and the difficulty in preserving garment details. The overall pipeline of our MagicTryOn is shown in Fig. 2(a). MagicTryOn takes videos of persons, clothing-agnostic masks, pose representations, and target garment images as input. Specifically, the videos of persons and pose representations are first encoded by the encoder into the latent space, producing the agnostic latent and pose latent, respectively. The clothing-agnostic masks are resized and mapped into the latent space to obtain the mask latent. These latents are then concatenated with random noise along the channel dimension to form the input for the DiT backbone. Meanwhile, we decompose the garment image into semantic, structural, and appearance streams. These streams are encoded by dedicated encoders to produce the text token, CLIP token, line token, and garment token. The text and CLIP tokens encode garment semantics, the line token encodes garment structure, and the garment token encodes garment appearance. The garment token is concatenated with the input token along the sequence dimension to provide global garment guidance. In addition, these tokens are fed into the DiT blocks to provide fine-grained garment detail conditioning. After multiple denoising steps through the DiT backbone, the network generates the latent representation of the try-on results, which is subsequently decoded into videos by the decoder.

## 3.1 Fine-grained Garment Preservation

Unlike generic video generation, the video virtual try-on task faces a unique challenge of maintaining garment pattern details and overall style under dynamic human poses and movements while ensuring natural, seamless visual coherence. Therefore, we propose a fine-grained garment-preservation strategy that decomposes the garment image into semantic, structural, and appearance

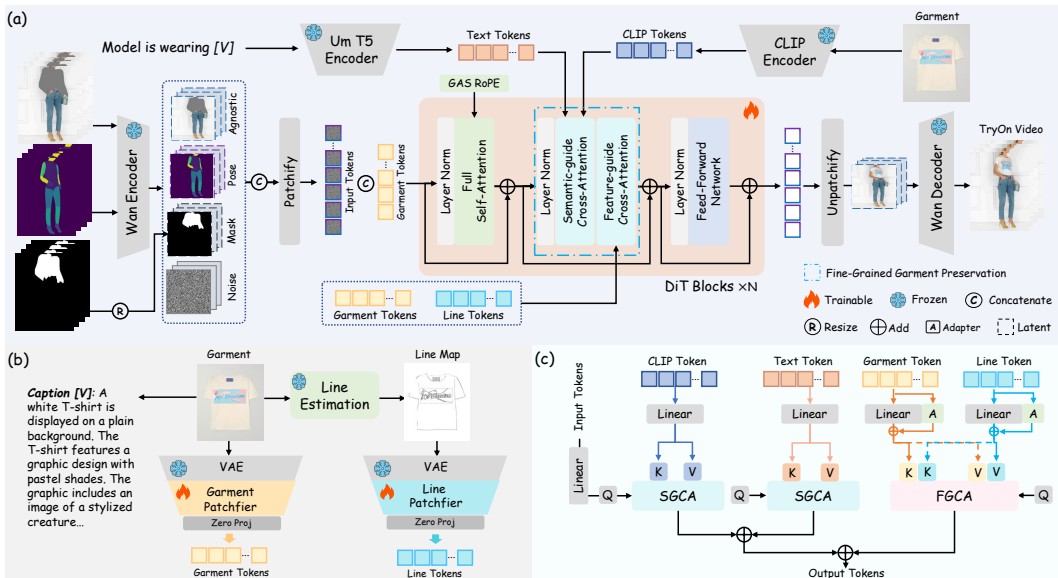

Figure 2: Overview of MagicTryOn. We introduce a garment-aware spatiotemporal RoPE within full self-attention to provide spatiotemporal modeling of garment features. We decompose the garment image into the text token and the CLIP token that encode semantics, the line token that encodes structure, and the garment token that encodes appearance, as shown in (b). Fine-grained garment preservation comprises semantic-guided cross-attention (SGCA) and feature-guided cross-attention (FGCA), as shown in (c). SGCA fuses semantic information, while FGCA fuses structural and appearance information to improve garment consistency.

cues and uses them to provide principled guidance during denoising, thereby improving garment consistency. In the following sections, we describe the garment-image decomposition pipeline and the subsequent injection of the decomposed tokens into the denoising network.

### 3.1.1 GARMENT INFORMATION DECOMPOSITION

We design a series of operations for decomposing garment details, as shown in Fig. 2(b). First, we introduce a line estimation module (Pan, 2025) to extract structure line maps from garment images, which encapsulate structural information and critical edges. Effectively leveraging these line maps provides stable structural guidance under dynamic human poses, enabling the network to better preserve the structural integrity of garments. Furthermore, we design a trainable Patchfier module subsequent to the frozen VAE encoder (Wang et al., 2025) to more effectively extract latent features from both garment images and line maps, obtaining garment token $\mathbf{T}_g$ and line token $\mathbf{T}_l$. Here, a zero projection is introduced to enhance training stability and mitigate potential latent collapse during the training process. Furthermore, we employ the Qwen2.5-VL-7B (Wang et al., 2024a) to generate highly-specific text descriptions of garment images, constructing a text vector $\mathbf{V}$ that encapsulates multiple attributes including color, style, and patterns. Notice that we use Qwen2.5-VL-7B to augment the existing try-on dataset (Dong et al., 2019; Choi et al., 2021; Morelli et al., 2022; Fang et al., 2024) by adding a caption attribute to each garment. *During inference, either the Qwen2.5-VL-7B generated caption or a user-provided caption can be used.* These generated descriptions are subsequently integrated with simplified prompts (*e.g.*, *Model is wearing [V]*) and fed into the UmT5 Encoder (Chung et al.) to derive the text token $\mathbf{T}_{txt}$. Meanwhile, semantic features of the garment images are extracted using a CLIP encoder (Radford et al., 2021), yielding the CLIP token $\mathbf{T}_{clip}$, as shown in Fig. 2(a).

### 3.1.2 INJECTION OF DECOMPOSED GARMENT INFORMATION

After obtaining the decomposed garment information, injecting it into the denoising network is a critical step. Simply concatenating these garment-related tokens along the sequence dimension exploits the information only coarsely, which results in losing fine-grained details during denoising. Therefore, we introduce Semantic-Guided Cross-Attention (SGCA) and Feature-Guided Cross-

Attention (FGCA) within DiT blocks to provide fine-grained garment detail guidance, as shown in Fig. 2(c). SGCA takes text tokens and CLIP tokens as inputs to supply global semantic representations of garments. In FGCA, we design a fine-grained control by keeping the query unchanged and concatenating the key and value of garment tokens and structure line tokens along the sequence dimension. This joint modeling enhances the model's ability to perceive and preserve complex garment details, thereby improving the fidelity and consistency of the generated results.

**SGCA.** We formally define the input token sequence as $\mathbf{T}_{\text{seq}}^{\text{in}} \in \mathbb{R}^{B \times L \times C}$. This sequence is projected as the query $\mathbf{Q}$, while CLIP tokens $\mathbf{T}_{\text{clip}}$ and text tokens $\mathbf{T}_{\text{txt}}$ are separately mapped to key-value pairs: $\mathbf{K}^{\text{clip}}$, $\mathbf{V}^{\text{clip}}$ and $\mathbf{K}^{\text{txt}}$, $\mathbf{V}^{\text{txt}}$. The decoupled cross-attention is performed as:

$$\text{SGCA}_i = \text{Attention}(\mathbf{Q}_i, \mathbf{K}_i^{\text{clip}}, \mathbf{V}_i^{\text{clip}}) + \text{Attention}(\mathbf{Q}_i, \mathbf{K}_i^{\text{txt}}, \mathbf{V}_i^{\text{txt}}). \tag{1}$$

Following the attention computation, we obtain fused output tokens $\mathbf{O}_s \in \mathbb{R}^{B \times L \times C}$ that consolidate garment-aware global semantics.

**FGCA.** We jointly incorporate both the garment token and the line token to perform cross-attention with the input token sequence. Specifically, we project $\mathbf{T}_{\text{seq}}^{\text{in}}$ into the query $\mathbf{Q}$, while the garment token $\mathbf{T}_{\text{g}}$ is projected into $\mathbf{K}^{\text{g}}$, $\mathbf{V}^{\text{g}}$, and the line token $\mathbf{T}_{\text{l}}$ is projected into $\mathbf{K}^{\text{l}}$, $\mathbf{V}^{\text{l}}$. The attention computations are then formulated as:

$$\text{FGCA}_i = \text{Attention}(\mathbf{Q}_i, [\mathbf{K}_i^{\text{g}}, \mathbf{K}_i^{\text{l}}], [\mathbf{V}_i^{\text{g}}, \mathbf{V}_i^{\text{l}}]), \tag{2}$$

where $[\cdot]$ means concatenated along the sequence dimension. After equation 2, we obtain detail-enriched output tokens $\mathbf{O}_t \in \mathbb{R}^{B \times L \times C}$. The final output sequence $\mathbf{T}_{\text{seq}}^{\text{out}} \in \mathbb{R}^{B \times L \times C}$ is obtained by adding $\mathbf{O}_s$ and $\mathbf{O}_t$. In FGCA, we propose a lightweight adapter module that facilitates efficient adaptation to the garment feature distribution during the fine-tuning of pretrained diffusion models, achieved by introducing only a small number of learnable parameters. This design not only improves the stability of the optimization process, but also enables more precise and fine-grained control over the generation of garment-related features.

### 3.2 GARMENT-AWARE SPATIOTEMPORAL RoPE

Maintaining a stable garment identity across frames remains challenging for video virtual try-on , as existing models lack spatiotemporal modeling tied to the garment itself. This limitation often manifests as temporal jitter and appearance drift under deformation. We address this by extending rotary position embedding (RoPE) to a garment-aware spatiotemporal (GAS) RoPE that encodes relative spatiotemporal relations for garment tokens, as shown in Fig. 2(a). Specifically, we concatenate garment token with input tokens along the sequence dimension $L$. Let $\mathbf{T}_{\text{inp}} \in \mathbb{R}^{B \times L \times C}$ denote the input tokens, where $B$ represents the batch size, $C$ denotes the channel dimension, and the sequence length $L = F \times H \times W$ (with $F$, $H$, and $W$ corresponding to the video's frames, height, and width, respectively). To incorporate garment information, we prepend a garment token of size $1 \times H \times W$ to the input sequence, thereby extending the sequence length to $L' = (F + 1) \times H \times W$. To ensure spatial positional encoding compatibility for the concatenated garment token, we adjust the grid size in RoPE computation from the original $[F, H, W]$ to $[F + 1, H, W]$. This modification enables both the garment token and input tokens to receive consistent positional encodings, allowing the denoising network to effectively recognize and utilize garment features. The concatenated sequence $\mathbf{T}_{\text{seq}} \in \mathbb{R}^{B \times L' \times C}$ is subsequently fed into a full self-attention module to capture inter-frame and image-garment dependency relationships.

For each position $p = (t, x, y)$ on the $[F + 1, H, W]$ grid with $t \in \{0, \dots, F\}$, $x \in \{0, \dots, H-1\}$, $y \in \{0, \dots, W-1\}$, (where $t = 0$ indexes the garment token), queries and keys are rotated before attention:

$$\tilde{\mathbf{q}}_{i,k}(p) = \mathbf{R}(\vartheta_k(p))\, \mathbf{q}_{i,k}(p), \qquad \tilde{\mathbf{k}}_{i,k}(p) = \mathbf{R}(\vartheta_k(p))\, \mathbf{k}_{i,k}(p), \tag{3}$$

where $\mathbf{R}(\theta) = \begin{bmatrix} \cos\theta & -\sin\theta \\ \sin\theta & \cos\theta \end{bmatrix}$ is a rotation applied to each pair of channels, $i$ indexes the head, $k$ indexes the channel pairs within a head, and the rotation angle is: $\vartheta_k(p) = \omega_k^t t + \omega_k^x x + \omega_k^y y$, where $\omega_k^{\{\cdot\}}$ are RoPE frequencies. The full self-attention is computed with the rotated queries/keys:

$$\text{Attention}_i(\mathbf{T}_{\text{seq}}) = \text{softmax}\left(\tilde{\mathbf{Q}}_i \tilde{\mathbf{K}}_i^\top d_h^{-1/2}\right) \mathbf{V}_i \quad \tilde{\mathbf{Q}}_i = \text{RoPE}(\mathbf{Q}_i), \ \tilde{\mathbf{K}}_i = \text{RoPE}(\mathbf{K}_i). \tag{4}$$

**Spatiotemporal Consistency Discussion.** Ensuring spatiotemporal consistency across frames is a key challenge in VVT task. Existing methods (Fang et al., 2024; He et al., 2024; Xu et al., 2024a; Li et al., 2025c; Nguyen et al., 2025) usually separate spatial and temporal attention, but this isolated design struggles to capture fine-grained spatiotemporal dependencies and dynamic changes, often leading to frame instability and garment flicker. To overcome this, we employ full self-attention that unifies spatial and temporal modeling, enabling interactions across all frames and positions to capture both intra-frame details and inter-frame dynamics. Moreover, we enhance this mechanism with a GAS RoPE and a prepended garment token, which jointly assign relative positions to garment and video tokens on a shared grid. This design provides reliable temporal anchors for garment features, strengthening cross-frame correspondence under deformation and reducing texture flicker.

## 3.3 TRAINING OBJECTIVE

As shown in Fig. 2, we conduct comprehensive fine-tuning of the DiT blocks and Patchfier modules based on pretrained weights, while keeping other modules frozen. During fine-tuning, in addition to using the standard diffusion loss, we introduce a mask-aware loss based on clothing-agnostic masks. This loss aims to enhance the network's focus and modeling capability on garment generation regions, thereby improving the restoration quality and consistency of garment details. The overall training objective is formulated as follows:

$$\mathcal{L} = \mathbb{E}_{t,\mathbf{x}_1,c,\mathbf{x}_0 \sim \mathcal{N}(0,\mathbf{I})} \left[ \|u(\mathbf{x}_t,t,c) - v_t\|^2 \right] + \mathbb{E}_{t,\mathbf{x}_1,c,\mathbf{x}_0 \sim \mathcal{N}(0,\mathbf{I})} \left[ \|\mathbf{M} \odot (u(\mathbf{x}_t,t,c) - v_t)\|^2 \right],$$
(5)

where $\mathbf{x}_1$ is video latent, $\mathbf{x}_0$ is a random noise, $\mathbf{x}_t$ represents a linear interpolation between $\mathbf{x}_0$ and $\mathbf{x}_1$. The groud truth velocity $v_t$ is: $v_t = \frac{dx_t}{dt} = x_1 - x_0$. $c$ is the condition, such as garment, text, pose, and agnostic. $u(\mathbf{x}_t,t,c)$ is the output velocity predicted by the model. $\mathbf{M}$ is the binary mask generated from the clothing-agnostic mask. $\odot$ denotes element-wise multiplication. *For the description of distribution-matching distillation, please refer to **Appendix A.2.***

## 4 EXPERIMENTS

### 4.1 DATASETS AND METRICS

We select two publicly available image virtual try-on datasets VITON-HD (Choi et al., 2021) and DressCode (Morelli et al., 2022) and one publicly available video try-on dataset ViViD (Fang et al., 2024) for hybrid training. Specifically, VITON-HD and DressCode contains 11,647 and 48,392 paired image training samples at 768×1024 resolution. ViViD includes 7,759 paired video training samples at 624×832 resolution. We evaluate our method on the test sets of ViViD (Fang et al., 2024) and VVT (Dong et al., 2019) for video virtual try-on. We conduct experiments under paired and unpaired settings. In the paired setting, the input garment matches the one worn by the human model, while in the unpaired setting, the model tries on a different garment. *During testing, the resolution of videos is the same as that in the original dataset.* We adopt four widely used metrics to evaluate the quality of video try-on results, including SSIM, LPIPS, VFID-I3D ($\text{VFID}_I$) (Fang et al., 2024), and VFID-ResNeXt ($\text{VFID}_R$) (Fang et al., 2024). VFID is used to evaluate both the spatial quality and temporal consistency of videos, where I3D (Carreira & Zisserman, 2017) and ResNeXt (Xie et al., 2017) are different backbone models. In the paired setting, all four metrics are used, whereas in the unpaired setting, only $\text{VFID}_I$ and $\text{VFID}_R$ are applied.

### 4.2 IMPLEMENTATION DETAILS

We adopt the pretrained weights from Wan2.1-Fun-Control (alibaba pai, 2025) as the foundational model, which is fine-tuned based on Wan2.1-I2V (Team, 2025). The model training employs a two-stage progressive strategy. In the first stage, we train the model using random image resolutions ranging from 256 to 512 on three datasets, including VITON-HD (Choi et al., 2021), DressCode (Morelli et al., 2022), and ViViD (Fang et al., 2024). During the second stage, training continues on the aforementioned datasets with image resolutions randomly sampled between 512 and 1024. For all stages, each training video sample contains 49 frames, with a batch size set to 2. The total number of training iterations is 45K (15K in stage one and 30K in stage two). The AdamW optimizer is utilized with a fixed learning rate of 1e-5. All training processes are conducted on 8 NVIDIA H20

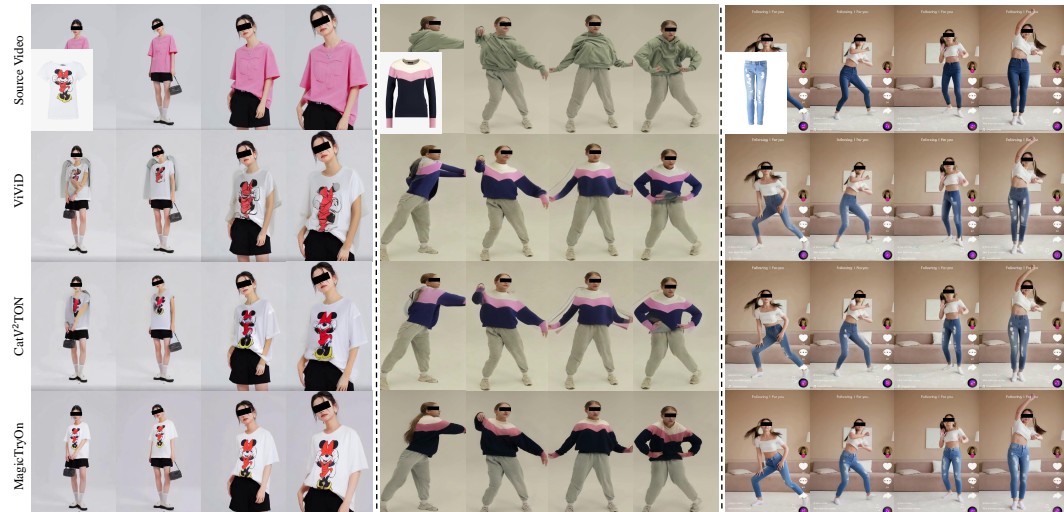

Figure 3: Qualitative comparison of video virtual try-on results under unconstrained settings, including model runway (*left*), complex and occluded motions (*middle*), and large-scale dance movements (*right*). The faces are blurred due to privacy concerns. Please **zoom-in** for better visualization.

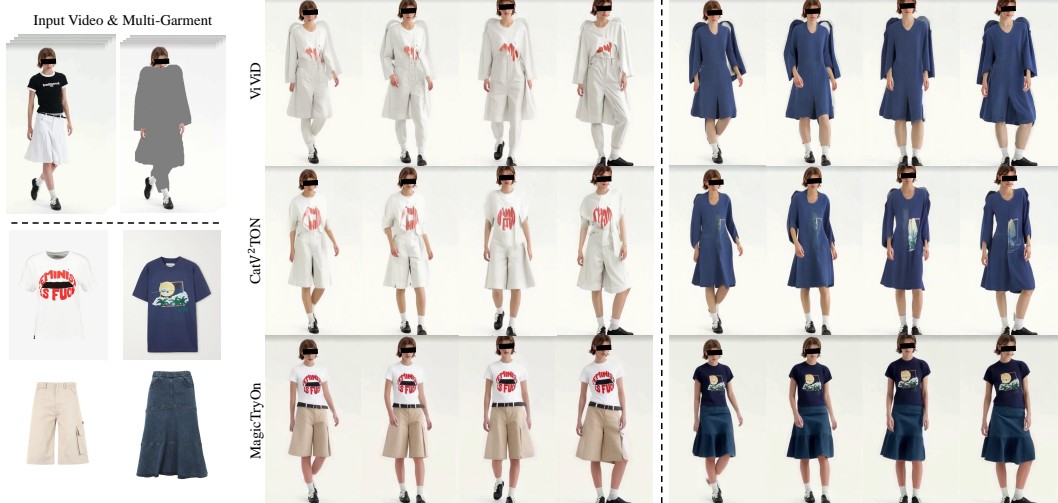

Figure 4: Qualitative comparison of video virtual try-on results on multi-garment scenarios. The faces are blurred due to privacy concerns. Please **zoom-in** for better visualization.

(96GB) GPUs. The number of inference steps during testing is set to 20. For the distilled version of MagicTryOn, the inference steps are reduced to 4. We call the distilled version MagicTryOn-Turbo.

## 4.3 COMPARISON WITH SOTA METHODS

In this section, we present comparisons on the ViViD (Fang et al., 2024) dataset between MagicTryOn and other methods, comparisons under unconstrained settings, as well as multi-garment scenarios. We also provide comparisons with image-based try-on methods (refer to ***Appendix A.4***), comparisons on the VVT (Dong et al., 2019) dataset (refer to ***Appendix A.9***), comparisons under in-the-wild settings (refer to ***Appendix A.10***), and a user study (refer to ***Appendix A.11***). ***We further provide video results in the Supplementary Material*** to better demonstrate the try-on performance.

**Comparison on the ViViD Dataset.** We conduct quantitative comparisons on the ViViD (Fang et al., 2024) datasets with state-of-the-art open-source video virtual try-on methods, as shown in Tab. 1. As can be seen, our method outperforms existing approaches across evaluation metrics, which demonstrates that our design can greatly enhance garment consistency and spatiotemporal stability

Figure 5: Qualitative comparison of cross-category try-on results, including trousers to shorts, trousers to skirts, and skirts to trousers. Please **zoom in** for details.

Table 1: Quantitative comparison on the ViViD (Fang et al., 2024) dataset. MagicTryOn-Turbo denotes the distilled version. The best results are in red. $p$ and $u$ denote the paired and unpaired settings, respectively. - indicates that video inference time and memory are not applicable to image-based try-on methods, or are not reported in the original paper.

| Methods | VFID$_I^p\downarrow$ | VFID$_R^p\downarrow$ | SSIM$\uparrow$ | LPIPS$\downarrow$ | VFID$_I^u\downarrow$ | VFID$_R^u\downarrow$ | GPU memory | Inference time |
|---|---|---|---|---|---|---|---|---|
| StableVITON (Kim et al., 2024) | 34.2446 | 0.7735 | 0.8019 | 0.1338 | 36.8985 | 0.9064 | - | - |
| OOTDiffusion (Xu et al., 2025) | 29.5253 | 3.9372 | 0.8087 | 0.1232 | 35.3170 | 5.7078 | - | - |
| IDM-VTON (Choi et al., 2024) | 20.0812 | 0.3674 | 0.8227 | 0.1163 | 25.4972 | 0.7167 | - | - |
| ViViD (Fang et al., 2024) | 17.2924 | 0.6209 | 0.8029 | 0.1221 | 21.8032 | 0.8212 | 62.59G | 204.183s |
| CatV$^2$TON (Chong et al., 2025) | 13.5962 | 0.2963 | 0.8727 | 0.0639 | 19.5131 | 0.5283 | 27.66G | 209.127s |
| DreamVVT (Zuo et al., 2025) | 11.0180 | 0.2549 | 0.8737 | 0.0619 | 16.9468 | 0.4285 | - | - |
| MagicTryOn-Turbo | 9.4082 | 0.2836 | 0.8745 | 0.0624 | 16.2213 | 0.4958 | 21.32G | **6.69s** |
| MagicTryOn | 8.4030 | 0.2346 | 0.9011 | 0.0602 | 14.7147 | 0.3200 | 51.51G | 345.271s |

in generated try-on videos. *Note that the inference time and GPU memory usage are measured on a single NVIDIA H20 when generating a 64-frame video at a resolution of 624×832.* We also provide qualitative comparisons on the ViViD dataset, please refer to ***Appendix A.5 and Fig. 8***.

**Comparison of the Distilled Version.** We compare the distilled version MagicTryOn-Turbo with other methods, as shown in Tab. 1. We observe that MagicTryOn-Turbo achieves comprehensive advantages over competing methods, especially in inference time, generating a 64-frame video with a resolution of 624×832 only takes 6.69s on a single H20 GPU. It is 30× faster than CatV$^2$TON while maintaining strong performance. Compared with MagicTryOn, it is 50× faster. Beyond its speed, MagicTryOn-Turbo also delivers strong try-on performance, showing substantial potential for practical deployment. Visual comparison results can be found in the ***Appendix A.6 and Fig. 9***.

**Comparison under Unconstrained Settings.** We compare our method with the open-source video virtual try-on methods ViViD (Fang et al., 2024) and CatV$^2$TON (Chong et al., 2025) under unconstrained settings, as shown in Fig. 3. ViViD and CatV$^2$TON suffer from garment texture blurring, detail loss, and inter-frame instability in runway, complex motion, and dance scenarios. In contrast, MagicTryOn maintains higher garment fidelity and stronger spatiotemporal consistency across all three scenarios, as it combines fine-grained garment feature modeling with spatiotemporal consistency enhancement, enabling the generation of more natural, realistic, and consistent try-on videos.

**Comparison in Multi-garment Scenarios.** We compare the performance of different methods on multi-garment try-on, as shown in Fig. 4. We observe that ViViD and CatV$^2$TON often produce incomplete or misaligned garment overlays in multi-garment scenarios, failing to preserve the patterns or colors of the second garment in some frames, which leads to unstable results. In contrast, our method not only clearly preserves the textures and patterns of multiple garments but also correctly maintains their compositional relationships, avoiding misalignment or blending errors. This superiority comes from our fine-grained garment feature disentanglement, which allows different garments to be modeled separately while maintaining their relative relationships, thereby preventing blurring and misalignment and generating more natural and consistent multi-garment video results.

**Comparison in Cross-Category Garment Scenarios.** We compare the performance of different methods under cross-category garment scenarios, as shown in Fig. 5. We observe that our method produces superior try-on results across various transformations, including skirt to trousers, trousers

Table 2: Quantitative comparisons on the ViViD (Fang et al., 2024) dataset, including Magic-TryOn versus Wan2.1-I2V (Team, 2025) and Wan2.1-Fun-Control (alibaba pai, 2025), as well as MagicTryOn-Hunyuan (using Hunyuan-DiT (Li et al., 2024) as the base model) versus CatV$^2$TON. $p$ and $u$ denote the paired and unpaired settings. The best and second-best results are in red and blue.

| Methods | VFID$_I^p$↓ | VFID$_R^p$↓ | SSIM↑ | LPIPS↓ | VFID$_I^u$↓ | VFID$_R^u$↓ | GPU memory | Inference time |
|---|---|---|---|---|---|---|---|---|
| Wan2.1-I2V (Team, 2025) | 18.6245 | 1.2303 | 0.7986 | 0.1786 | 22.2147 | 1.0087 | 56.55G | 359.634s |
| Wan2.1-Fun-Control (alibaba pai, 2025) | 14.2180 | 0.7113 | 0.8529 | 0.0818 | 19.9284 | 0.8656 | 54.68G | 356.686s |
| CatV$^2$TON (Chong et al., 2025) | 13.5962 | 0.2963 | 0.8727 | 0.0639 | 19.5131 | 0.5283 | 27.66G | 209.127s |
| MagicTryOn-Hunyuan | 10.1835 | 0.2782 | 0.8956 | 0.0607 | 15.6360 | 0.3209 | 20.02G | 205.106s |
| MagicTryOn | 8.4030 | 0.2346 | 0.9011 | 0.0602 | 14.7147 | 0.3200 | 51.51G | 345.271s |

Table 3: Ablation study of each component on the ViViD (Fang et al., 2024) test set with a resolution of 624×832 and 64 frames. $p$ and $u$ denote the paired and unpaired settings, respectively.

| Metric | w/o GAS | w/o SGCA-T | w/o SGCA-C | w/o SGCA | w/o FGCA-G | w/o FGCA-L | w/o FGCA | w/o mask | Full model |
|---|---|---|---|---|---|---|---|---|---|
| VFID$_I^p$↓ | 16.1083 | 18.6721 | 16.0452 | 19.2796 | 17.4817 | 16.4579 | 17.7598 | 18.3322 | 12.1988 |
| VFID$_R^p$↓ | 0.5080 | 0.7971 | 0.5447 | 0.9075 | 0.8284 | 0.7182 | 0.9304 | 0.5147 | 0.2346 |
| SSIM↑ | 0.8429 | 0.8832 | 0.8535 | 0.8163 | 0.8683 | 0.8619 | 0.8511 | 0.8458 | 0.8841 |
| LPIPS↓ | 0.0953 | 0.0830 | 0.0862 | 0.0884 | 0.0870 | 0.0833 | 0.0882 | 0.1057 | 0.0815 |
| VFID$_I^u$↓ | 23.2657 | 24.6428 | 24.6383 | 25.1229 | 25.2449 | 23.6495 | 25.6789 | 24.5531 | 17.5710 |
| VFID$_R^u$↓ | 0.8544 | 0.8128 | 0.8283 | 0.9247 | 0.9324 | 0.8704 | 1.0106 | 0.9260 | 0.5073 |

to shorts, and trousers to skirts. This demonstrates that MagicTryOn is not constrained by the shape of the input mask. MagicTryOn can generate garment contours and structures that match the target clothing, without being restricted by the mask shape of the original garment.

**Effectiveness Beyond the Base Model.** To verify that the improvement in try-on performance primarily stems from our module design, we compare MagicTryOn with the video backbones Wan2.1 (Team, 2025). For fairness, we fine-tune Wan2.1-I2V (Team, 2025) and Wan2.1-Fun-Control (alibaba pai, 2025) on the same try-on datasets and use Qwen2.5-VL-7B (Wang et al., 2024a) for garment captioning in both settings. As shown in Tab. 2, merely fine-tuning the Wan2.1 backbones fails to achieve optimal try-on performance. In contrast, introducing our proposed modules on the same backbone yields the best results. This indicates that the performance gains are primarily attributable to our architectural design rather than the base model itself. Qualitative comparisons are provided in ***Appendix A.7 and Fig. 10.***

To further show that the gains are mainly attributable to our proposed strategies and modules rather than the base model, we conduct a controlled study using Hunyuan-DiT (Li et al., 2024) as the backbone. Specifically, we integrate garment-aware spatiotemporal RoPE, fine-grained garment preservation, and a mask-aware loss into Hunyuan-DiT, and train MagicTryOn-Hunyuan under the same experimental settings as in Section 4.2. We compare it with CatV$^2$TON (Chong et al., 2025), which also uses Hunyuan-DiT as the backbone, as shown in Tab. 2. The results show that MagicTryOn-Hunyuan outperforms CatV$^2$TON across all metrics, indicating that the performance improvements are primarily attributable to our modular design rather than the inherent capability of the base model. Corresponding qualitative comparisons are provided in ***Appendix A.8 and Fig. 11.***

## 4.4 Ablation Study

To evaluate each component's contribution to overall performance, we conduct ablation studies on the fine-grained garment-preservation strategy, garment-aware spatiotemporal RoPE, and the mask-aware loss. All variants are trained for a total of 25K iterations (15K in stage one and 10K in stage two) using the same datasets as in Section 4.1. Quantitative results for each variant are shown in Tab. 3. We also provide visual comparisons of the ablation variants in ***Appendix A.15 and Fig. 14***.

**Fine-grained garment preservation.** We design six variants to perform ablation studies on the SGCA and FGCA modules in the fine-grained garment preservation strategy. Specifically, for the SGCA module, we construct three variants, removing the text token branch (*w/o* SGCA-T), removing the CLIP token branch (*w/o* SGCA-C), and completely removing the SGCA module (*w/o* SGCA). For the FGCA module, we adopt the same settings, obtaining the three variants: *w/o* FGCA-G, *w/o* FGCA-L, and *w/o* FGCA. The results are shown in Tab. 3. As can be seen, each garment-related token contributes positively to the network's generation performance, and the absence of these tokens significantly degrades the generated results. This demonstrates that injecting various

Table 4: Additive study of each component on the ViViD (Fang et al., 2024) test set with a resolution of 384×512 and 64 frames. $p$ and $u$ denote the paired and unpaired settings, respectively.

| Variants | VFID$_I^p$↓ | VFID$_R^p$↓ | SSIM↑ | LPIPS↓ | VFID$_I^u$↓ | VFID$_R^u$↓ |
|---|---|---|---|---|---|---|
| Bare Model | 21.9270 | 1.2376 | 0.8087 | 0.1181 | 28.2298 | 1.2003 |
| + SGCA | 19.2988 | 0.9313 | 0.8329 | 0.1007 | 25.3479 | 1.0360 |
| + SGCA + FGCA | 18.0794 | 0.7001 | 0.8630 | 0.0839 | 23.3136 | 0.8925 |
| + SGCA + FGCA + mask loss | 15.4081 | 0.5252 | 0.8704 | 0.0791 | 20.2338 | 0.7249 |
| + SGCA + FGCA + mask loss + GAS RoPE | 12.0640 | 0.2019 | 0.8852 | 0.0747 | 18.0523 | 0.5068 |

garment-related information into the denoising network through fine-grained garment preservation is effective and essential for maintaining both structural and semantic fidelity.

**GAS RoPE.** To verify the effectiveness of the garment-aware spatiotemporal (GAS) RoPE, we design a variant that removes GAS RoPE, referred to as *w/o GAS*. As shown in Tab. 3, removing GAS RoPE degrades generation performance. Without GAS RoPE, the network cannot assign garment-aware relative positions, leading to inaccurate preservation of garment style and noticeable temporal jitter. This indicates that GAS RoPE provides preliminary garment-structure anchors during denoising, which are crucial for maintaining overall style and cross-frame consistency.

**Mask-aware Loss.** To validate the role of the mask-aware loss, we design a variant that does not utilize the mask-aware loss during training, referred to as *w/o mask*, as shown in Tab. 3. We notice that removing the mask-aware loss leads to a degradation in overall model performance. This indicates that the mask-aware loss effectively guides the model to focus on and optimize clothing areas, thereby enhancing the accuracy and coherence of the generated results.

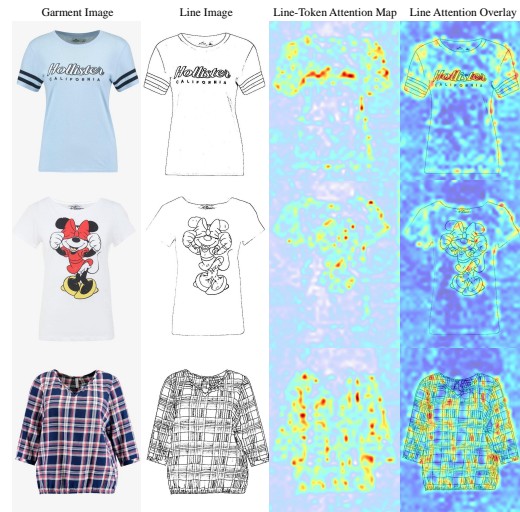

Figure 6: Visualizations of the line-token attention map. The attention weights are primarily concentrated on key garment regions, such as sharp details, patterns, and logos.

**Additive of Incremental Components.** We supplement additive experiments that progressively incorporate each module into the bare model to more clearly demonstrate the marginal contribution of every component. Specifically, starting from the Bare Model, we gradually add the SGCA module, the FGCA module, the mask loss, and the GAS RoPE, as shown in Tab. 4. As shown, each incremental component consistently improves performance, with clear and steady gains across all metrics as the SGCA, FGCA, mask loss, and GAS modules are progressively incorporated.

**Line-Token Attention Map.** To better demonstrate the contribution of the decomposed structural cues, we provide visualizations of the line-token attention map, as shown in Fig. 6. We observe that the attention weights are primarily concentrated on key garment regions, such as sharp details, patterns, and logos. This helps the model better understand fine-grained structural cues of garments and produce more accurate try-on results.

## 5 CONCLUSION

In this paper, we present MagicTryOn, a diffusion-transformer framework for garment-preserving video virtual try-on. Our system integrates a fine-grained garment-preservation module that decomposes garment cues and injects them via cross-attention, a garment-aware spatiotemporal RoPE to stabilize cross-frame identity, and a mask-aware loss to enhance fidelity in garment regions. Additionally, distribution-matching distillation compresses inference to 4 steps (50× faster). These components deliver superior garment-detail fidelity and temporal stability, and extensive experiments demonstrate state-of-the-art performance in unconstrained settings.

## ETHICS STATEMENT

This work relies on publicly available datasets under their respective licenses. No new data involving human subjects were collected. All visualizations respect privacy. We confirm that our method and experiments do not raise additional ethical concerns.

## REPRODUCIBILITY STATEMENT

We use publicly accessible datasets, VITON-HD (Choi et al., 2021), DressCode (Morelli et al., 2022), ViViD (Fang et al., 2024), and VVT (Dong et al., 2019). After the blind review period, we will release our codebase, training/inference scripts, configuration files, and model checkpoints, together with step-by-step instructions and evaluation protocols to fully reproduce all results.

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

# A APPENDIX

In the appendix, we first provide the LLM usage statement. We then present a detailed description of distribution-matching distillation. Subsequently, we report additional experimental comparisons, including image-based try-on baselines and a series of visual evaluations on multiple datasets and backbones. We also provide the user study setup and results, followed by visual outcomes from our ablation studies.

CONTENTS

## A.1 USE OF LLMS

The LLMs are used only for language polishing and editing of the manuscript text, primarily to refine grammar and word choice in the Introduction and Related Work sections.

## A.2 DISTRIBUTION-MATCHING DISTILLATION

Given a strong bidirectional teacher diffusion model (teacher) and a causal few-step student generator (student), our goal is to make the student's conditional distributions at key timesteps align with the teacher's under a budget of just four sampling steps, thereby markedly reducing latency while preserving garment fidelity and temporal stability.

### A.2.1 STEP 1: ODE INITIALIZATION

Training the causal few-step student directly with the distribution matching distillation (DMD) loss tends to be unstable due to architectural and information-flow mismatches with the bidirectional teacher. To address this, we first construct a small set of deterministic ODE trajectories using the teacher and perform an efficient regression-based initialization of the student, which significantly stabilizes the subsequent distillation (Yin et al., 2025). We use MagicTryOn to generate the ODE data. We construct the ODE data as follows.

First, sample a set of initial latent variables from a standard Gaussian:

$$\{x_T^{(i)}\}_{i=1}^L \sim \mathcal{N}(0, I). \tag{6}$$

Sxecond, use the pretrained bidirectional teacher with an ODE solver to deterministically simulate the reverse process from $T$ to 0:

$$\left\{x_t^{(i)}\right\}_{t=T \to 0,} \quad i = 1, \dots, L. \tag{7}$$

producing full trajectories over the teacher's 50-step schedule. Third, select the four student-aligned timesteps $S = \{0, 36, 44, 49\}$ from each trajectory and cache the corresponding states $\left\{x_{t_k}^{(i)}\right\}_{k=1}^4$ and their targets $\left\{x_0^{(i)}\right\}_{i=1}^L$. For initialization, we run a brief regression phase so that the student generator $G_\phi$ learns a few-step mapping to $x_0$ using the following loss:

$$\mathcal{L}_{\text{init}} = \mathbb{E}_{\{x_{t_i}\}, \{t_i\}} \left\| G_\phi\big(\{x_{t_i}^{(i)}\}_{i=1}^N, \{t_i\}_{i=1}^N\big) - \{x_0^{(i)}\}_{i=1}^N \right\|_2^2. \tag{8}$$

### A.2.2 STEP 2: DISTRIBUTION-MATCHING DISTILLATION

Distribution matching distillation converts a slow, multi-step teacher diffusion model into an efficient few-step student generator by minimizing a reverse KL divergence across randomly sampled timesteps $t$. Concretely, we match the student's output distribution $p_{\text{gen},t}(x_t)$ to the teacher-smoothed data distribution $p_{\text{data},t}(x_t)$ (obtained via the teacher diffusion process).

$$\mathcal{L}_{\text{DMD}} \triangleq \mathbb{E}_t \left[ \text{KL}\big(p_{\text{gen},t} \| p_{\text{data},t}\big) \right]. \tag{9}$$

The gradient of the reverse KL can be approximated by the difference of score functions evaluated along the student's sample path:

$$\nabla_\phi \mathcal{L}_{\text{DMD}} \triangleq \mathbb{E}_t \left[ \nabla_\phi \, \text{KL}\big(p_{\text{gen},t} \| p_{\text{data},t}\big) \right] \tag{10}$$

$$\approx -\mathbb{E}_t \left( \int \left[ s_{\text{data}}\big(\Psi(G_\phi(\epsilon), t), t\big) - s_{\text{gen},\xi}\big(\Psi(G_\phi(\epsilon), t), t\big) \right] \frac{d\, G_\phi(\epsilon)}{d\phi} \, d\epsilon \right). \tag{11}$$

Here, $\Psi$ denotes the forward diffusion operator that maps a clean sample to its noised version at time $t$. $G_\phi$ is the few-step generator (student) parameterized by $\phi$. $\epsilon \sim \mathcal{N}(0, I)$ is Gaussian noise. $s_{\text{data}}(x_t, t) = \nabla_{x_t} \log p_{\text{data},t}(x_t)$ is the data score (approximated by the pretrained teacher network). $s_{\text{gen},\xi}(x_t, t) = \nabla_{x_t} \log p_{\text{gen},t}(x_t)$ is the generator score (given by the student). During training, DMD (Yin et al., 2024) initializes both score functions using a pretrained diffusion model. The data score is kept fixed, whereas the generator score is updated online from the generator's current outputs. In parallel, the generator itself is optimized to move its output distribution toward the data distribution.

After training, the student model performs four-step inference, substantially reducing computational complexity and runtime while meeting real-time requirements without sacrificing garment detail and temporal stability.

### A.2.3 IMPLEMENTATION DETAILS

During training, we first use MagicTryOn to generate 6K ODE pairs from the ViViD (Fang et al., 2024) dataset and use them to initialize the student model, training for 6K iterations with AdamW at a learning rate of $5 \times 10^{-6}$. We then switch to the DMD objective and continue training for 12K iterations with AdamW at a learning rate of $2 \times 10^{-6}$. For additional details about DMD, please refer to CausVid (Yin et al., 2025).

Table 5: Quantitative comparison under multi-garment scenarios. The best results are in red.

| Methods | $\text{VFID}_I^u\downarrow$ | $\text{VFID}_R^u\downarrow$ |
|---|---|---|
| CatV$^2$TON | 61.6164 | 14.1268 |
| Ours | 26.1804 | 5.6258 |

## A.3 QUANTITATIVE METRICS UNDER MULTI-GARMENT SCENARIOS

We conduct a quantitative comparison with CatV$^2$TON (Chong et al., 2025) in the multi-garment scenario. Since no paired data are available in this setting, we compute the $VFID_I^u$ and $VFID_R^u$ metrics, as shown in Tab. 5. Combining the quantitative metrics and visual comparisons, Magic-TryOn outperforms existing methods in the multi-garment scenario.

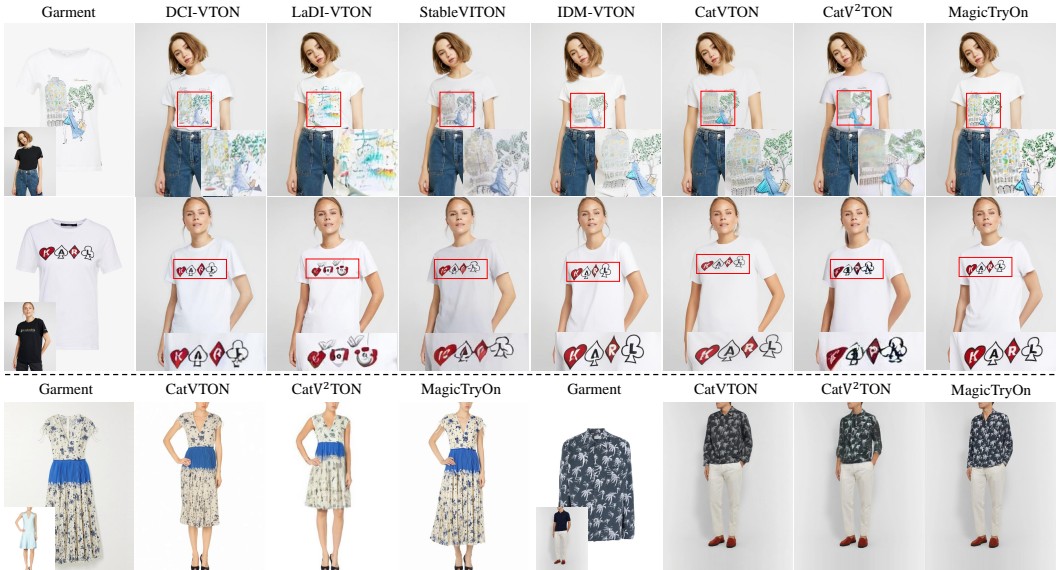

Figure 7: Qualitative comparison of image virtual try-on results on the VITON-HD (Choi et al., 2021) (*1-st* and *2-nd* **row**) and DressCode (Morelli et al., 2022) (*3-rd* **row**) datasets. Please **zoom-in** for better visualization.

Table 6: Quantitative comparison with other methods on image virtual try-on datasets. The best and second-best results are in red and blue. $p$ and $u$ denote the paired setting and unpaired setting.

|  | Metric | Methods | | | | | | |
|---|---|---|---|---|---|---|---|---|
|  |  | GP-VTON | LaDI-VTON | IDM-VTON | OOTDiffusion | CatVTON | CatV$^2$TON | MagicTryOn |
| VITON-HD | FID$^p$ ↓ | 8.726 | 11.386 | 6.338 | 9.305 | 6.139 | 8.095 | 4.959 |
|  | KID$^p$ ↓ | 3.944 | 7.248 | 1.322 | 4.086 | 0.964 | 2.245 | 0.572 |
|  | SSIM ↑ | 0.8701 | 0.8603 | 0.8806 | 0.8187 | 0.8691 | 0.8902 | 0.9104 |
|  | LPIPS ↓ | 0.0585 | 0.0733 | 0.0789 | 0.0876 | 0.0973 | 0.0572 | 0.0429 |
|  | FID$^u$ ↓ | 11.844 | 14.648 | 9.611 | 12.408 | 9.143 | 11.222 | 9.079 |
|  | KID$^u$ ↓ | 4.310 | 8.754 | 1.639 | 4.689 | 1.267 | 2.986 | 1.032 |
| DressCode | FID$^p$ ↓ | 9.927 | 9.555 | 6.821 | 4.610 | 3.992 | 5.722 | 6.550 |
|  | KID$^p$ ↓ | 4.610 | 4.683 | 2.924 | 0.955 | 0.818 | 2.338 | 0.725 |
|  | SSIM ↑ | 0.7711 | 0.7656 | 0.8797 | 0.8854 | 0.8922 | 0.9222 | 0.9295 |
|  | LPIPS ↓ | 0.1801 | 0.2366 | 0.0563 | 0.0533 | 0.0455 | 0.0367 | 0.0301 |
|  | FID$^u$ ↓ | 12.791 | 10.676 | 9.546 | 12.567 | 6.137 | 8.627 | 11.727 |
|  | KID$^u$ ↓ | 6.627 | 5.787 | 4.320 | 6.627 | 1.549 | 3.838 | 1.544 |

## A.4 COMPARISON WITH IMAGE-BASED TRY-ON METHODS

For image virtual try-on benchmarking, we conduct evaluations on the test sets of VITON-HD (Choi et al., 2021) and DressCode (Morelli et al., 2022). The testing experiments are conducted under two settings, paired and unpaired. In the paired setting, the input garment image and the garment worn by the human model are the same item. In contrast, the human model tries on different garment in the unpaired setting. During testing, the resolution of images is the same as that in the original dataset. We adopt four widely used metrics to evaluate the quality of image try-on results, including SSIM, LPIPS, FID, and KID. SSIM and LPIPS measure the similarity between two individual images, while FID and KID evaluate the similarity between two image distributions. In the paired setting, all four metrics are used, whereas in the unpaired setting, only FID and KID are applied.

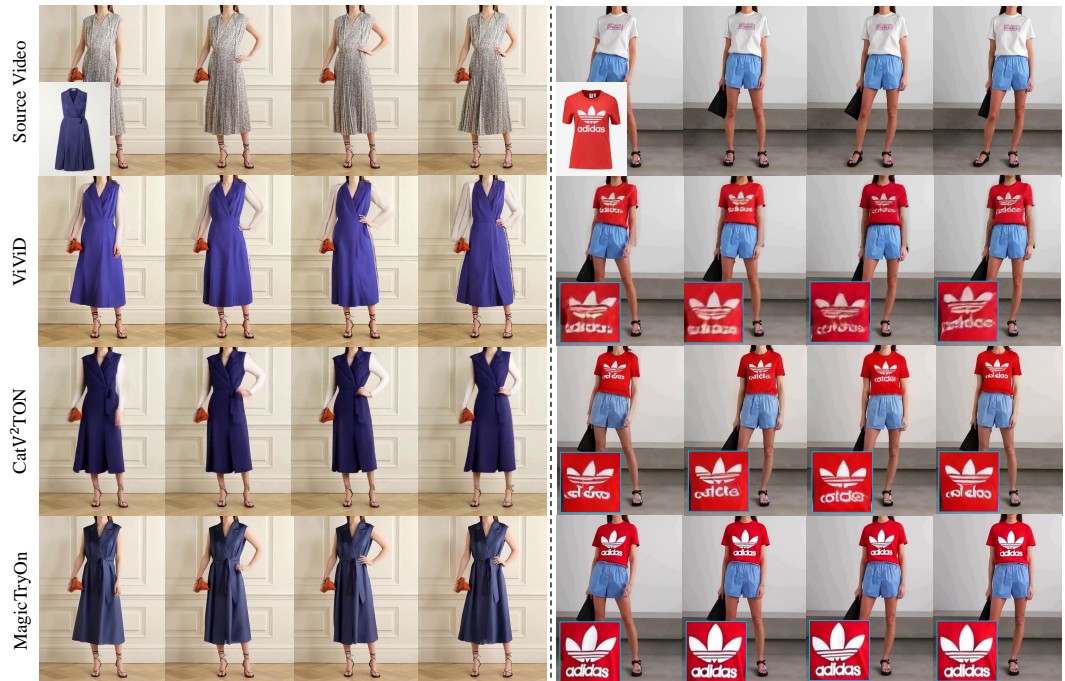

Figure 8: Qualitative comparison of video virtual try-on results on the ViViD (Fang et al., 2024) dataset. Please **zoom-in** for better visualization.

We perform quantitative comparisons with state-of-the-art image-based methods on the VITON-HD (Choi et al., 2021) and DressCode (Morelli et al., 2022) datasets under both paired and unpaired settings. As shown in Tab. 6, our method consistently outperforms existing approaches across multiple metrics, particularly in the unpaired scenario. This demonstrates that the use of full self-attention not only facilitates temporal consistency modeling but also enhances spatial perception, further highlighting the effectiveness of the proposed coarse-to-fine garment preservation strategy. Fig. 7 presents the visual results of different methods on the image virtual try-on task. As can be observed, our method demonstrates superior performance in preserving complex garment patterns compared to other methods specifically designed for image try-on.

A.5 VISUAL COMPARISON ON THE VIVID DATASET

Fig. 8 shows the qualitative comparison between our method and existing open-source video virtual try-on approaches. We observe that our method achieves outstanding performance in generating try-on videos, with improved temporal coherence and garment consistency—including color, style, and pattern. The garments also exhibit natural wrinkles and motion in response to human movement, demonstrating effective spatiotemporal modeling and fine detail preservation.

A.6 VISUAL COMPARISON OF THE DISTILLED VERSION

Fig. 9 presents visual comparisons of the distilled MagicTryOn-Turbo against multiple methods on the ViViD (Fang et al., 2024) dataset. With only four inference steps, MagicTryOn-Turbo still performs stable and accurate garment transfer. In terms of detail and style fidelity, high-frequency patterns—such as logos, stripes, and lettering—remain sharp and faithful. In terms of spatiotemporal consistency, the garment appearance varies smoothly with human motion and remains stable across frames. Despite its very high speed, MagicTryOn-Turbo maintains high try-on quality and strong spatiotemporal consistency, meeting real-time requirements. These results indicate that our distribution-matching distillation achieves substantial step reduction without sacrificing garment detail or stability.

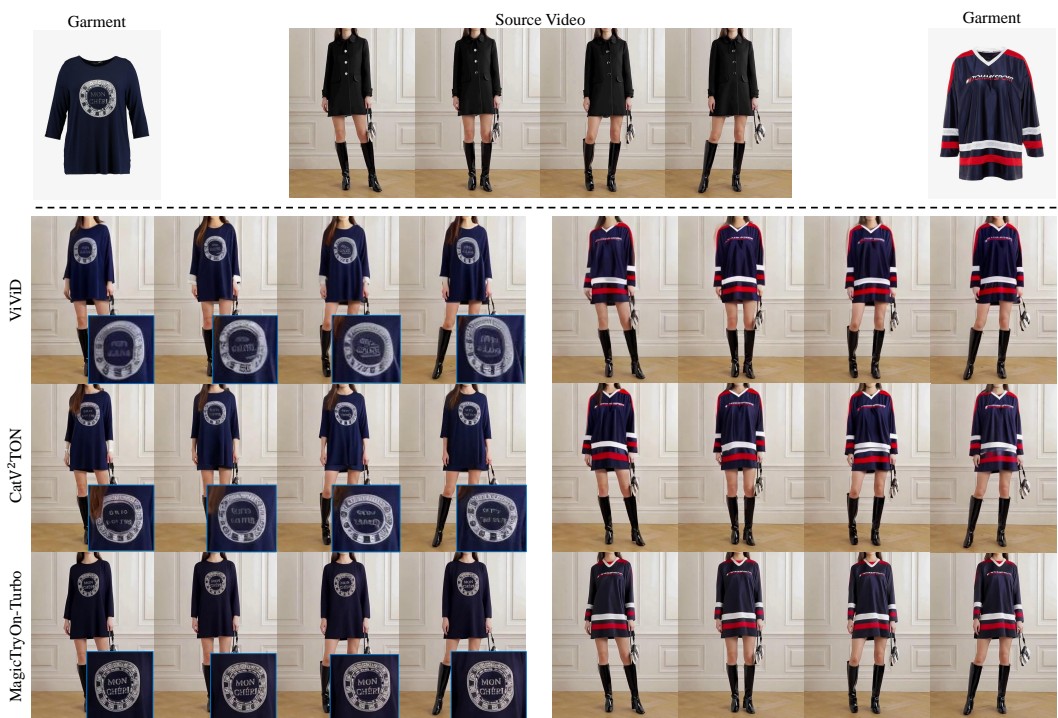

Figure 9: Visual comparison of MagicTryOn-Turbo and other methods on the ViViD (Fang et al., 2024) dataset. Please **zoom-in** for better visualization.

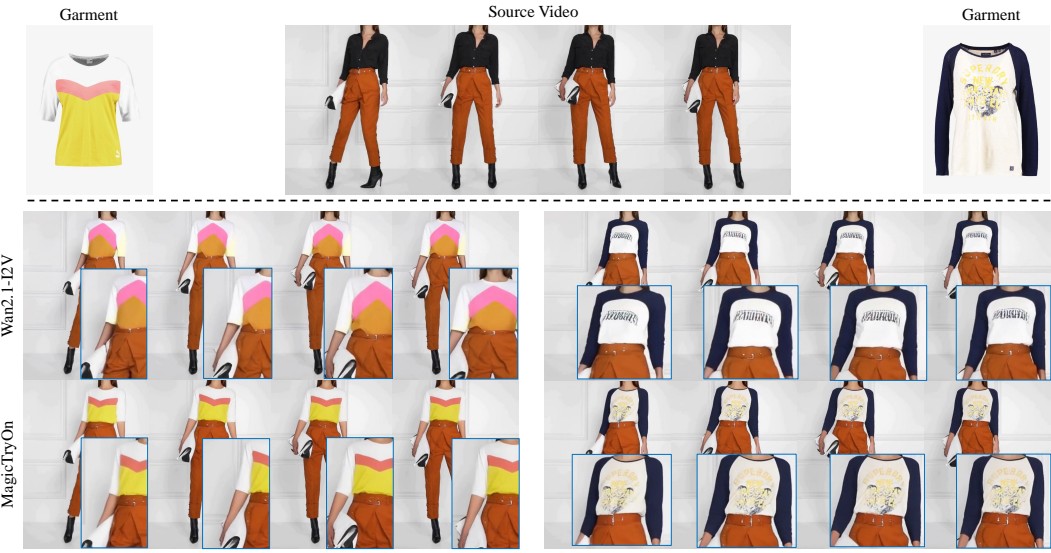

Figure 10: Visual comparison with the base model Wan2.1-I2V (Team, 2025) under ViViD (Fang et al., 2024) dataset. Please **zoom-in** for better visualization.

## A.7 VISUAL COMPARISON WITH WAN2.1

Fig. 10 compares the results of the video backbone Wan2.1-I2V (Team, 2025) and MagicTryOn under identical inputs and the same captioning, with captions generated by Qwen2.5-VL (Wang et al., 2024a). With the backbone alone, Wan2.1-I2V exhibits incomplete transfer of garment patterns and textures and weaker garment consistency. In contrast, MagicTryOn better preserves overall garment

Table 7: Quantitative comparison on the VVT (Dong et al., 2019) dataset. The best and second-best results are in red and blue. $p$ and $u$ denote the paired and unpaired settings, respectively.

| Methods | $\text{VFID}^p_I\downarrow$ | $\text{VFID}^p_R\downarrow$ | SSIM↑ | LPIPS↓ | $\text{VFID}^u_I\downarrow$ | $\text{VFID}^u_R\downarrow$ |
|---|---|---|---|---|---|---|
| FW-GAN (Dong et al., 2019) | 8.019 | 0.1215 | 0.675 | 0.283 | - | - |
| MV-TON (Deng et al., 2023) | 8.367 | 0.0972 | 0.853 | 0.233 | - | - |
| ClothFormer (Jiang et al., 2022) | 3.967 | 0.0505 | 0.921 | 0.081 | - | - |
| ViViD (Fang et al., 2024) | 3.793 | 0.0348 | 0.822 | 0.107 | 3.994 | 0.0416 |
| SwiftTry (Nguyen et al., 2025) | - | - | 0.887 | 0.066 | 3.589 | 0.5340 |
| CatV$^2$TON (Chong et al., 2025) | 1.778 | 0.0103 | 0.900 | 0.039 | 1.902 | 0.0141 |
| MagicTryOn-Hunyuan | 1.690 | 0.0097 | 0.902 | 0.038 | 1.834 | 0.0125 |
| MagicTryOn | 1.487 | 0.0039 | 0.917 | 0.024 | 1.662 | 0.0053 |

style and fine textures, and achieves stronger cross-frame consistency. Since both methods share the same backbone and the same captioning pipeline, these gains are attributable to our proposed modules: the fine-grained garment-preservation strategy, the garment-aware spatiotemporal RoPE, and the mask-aware loss, rather than to the base model's capacity.

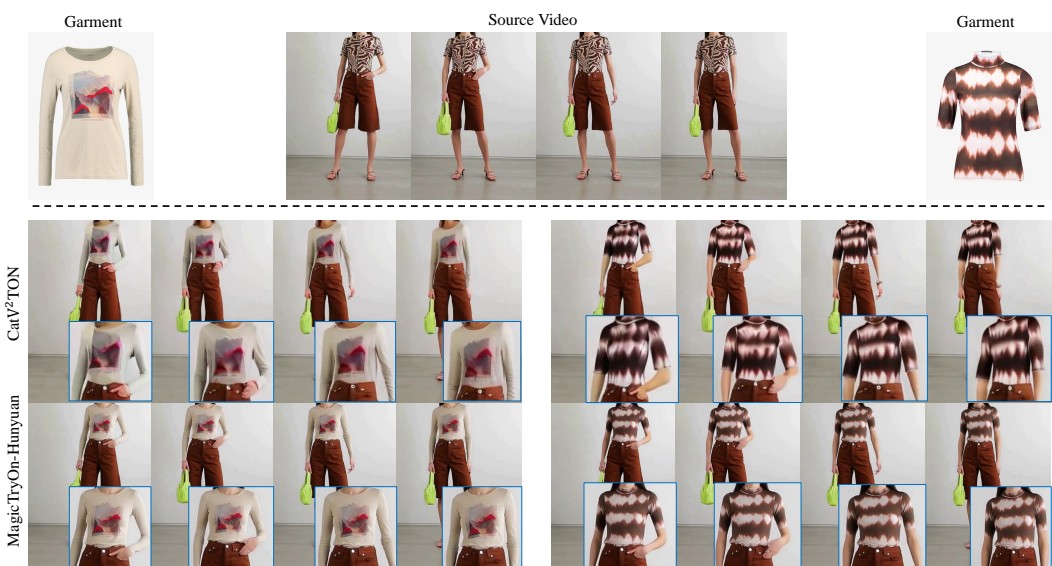

Figure 11: Qualitative comparison between our design and CatV$^2$TON (Chong et al., 2025) under the same base model (Hunyuan-DiT (Li et al., 2024)) on the ViViD (Fang et al., 2024) dataset. Please **zoom-in** for better visualization.

## A.8 Visual Comparison using Hunyuan-DiT

Fig. 11 compares CatV$^2$TON (Chong et al., 2025) and MagicTryOn-Hunyuan under the same backbone Hunyuan-DiT (Li et al., 2024). CatV$^2$TON shows deficiencies in pattern reconstruction and boundary stability, texture details tend to deform, and consistency around the neckline is weaker. In contrast, MagicTryOn-Hunyuan better preserves the overall garment style and high-frequency details (e.g., patterns, stripes, lettering) and maintains stronger spatiotemporal consistency across frames. Because both methods share the Hunyuan-DiT backbone, these improvements can be attributed to our module design, rather than to differences in the base model.

## A.9 Visual Comparison on the VVT Dataset

Tab. 7 reports quantitative results on the VVT (Dong et al., 2019) dataset. We observe that Magic-TryOn attains the best performance across all metrics, achieving the lowest VFID in both paired and

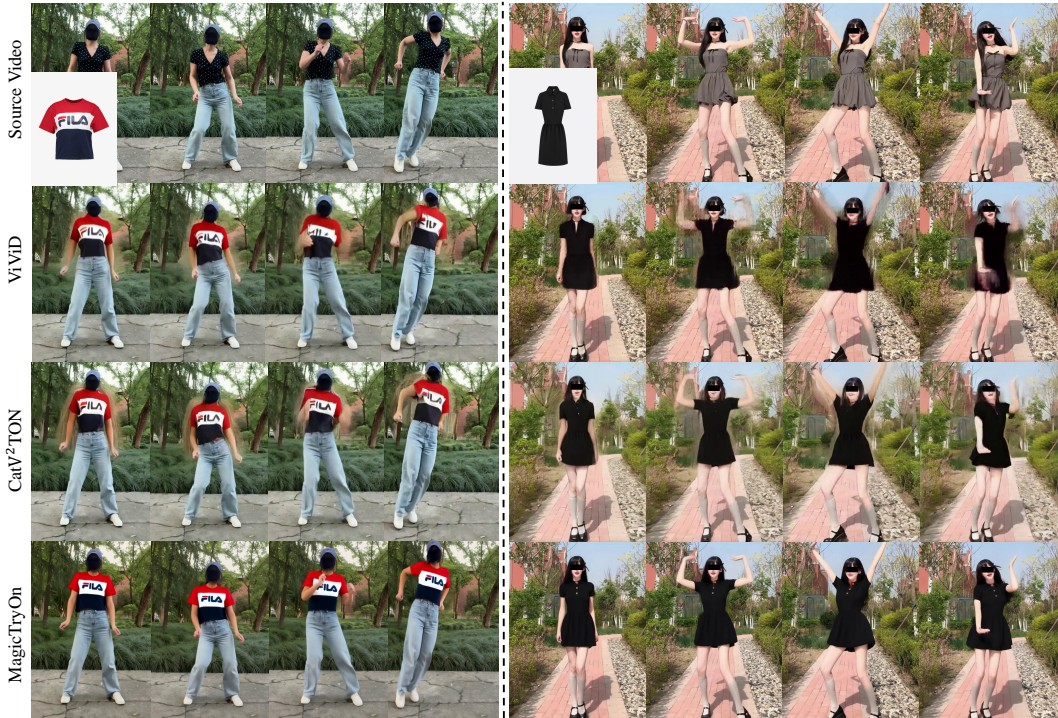

Figure 12: Visual comparison of different methods in in-the-wild scenarios. We select two dance cases to evaluate garment consistency and spatiotemporal stability. The faces are blurred due to privacy concerns. Please **zoom-in** for better visualization.

unpaired settings, as well as the highest SSIM and the lowest LPIPS. MagicTryOn-Hunyuan ranks second on every metric, surpassing all prior methods including CatV$^2$TON (Chong et al., 2025). Because lower VFID/LPIPS and higher SSIM indicate better perceptual quality and structural fidelity, these results demonstrate that our approach delivers superior garment fidelity and spatiotemporal stability on the VVT dataset.

## A.10 VISUAL COMPARISON IN IN-THE-WILD SCENARIOS

Fig. 12 compares different methods on two dance videos in in-the-wild scenarios. Existing methods generally struggle to preserve garment content and exhibit temporal instability: patterns and lettering drift or stretch under rapid motion, colors and textures vary randomly over time, flicker is noticeable, and cross-frame consistency is weak. In contrast, MagicTryOn delivers higher garment fidelity and stronger spatiotemporal stability in both dance cases. This advantage arises from our fine-grained garment-preservation strategy and the garment-aware spatiotemporal RoPE, which explicitly constrains cross-frame correspondences, jointly improving garment consistency and stability in complex in-the-wild motion.

## A.11 USER STUDY

We conduct a user study involving 50 participants to evaluate the performance of our method in comparison to CatV$^2$TON from two perspectives: temporal consistency and garment consistency. Each participant views 8 try-on video pairs for each aspect (a total of 16 questions) and is asked to select the more favorable result. The 8 generated try-on videos spanned diverse in-the-wild scenarios. The visual results are shown

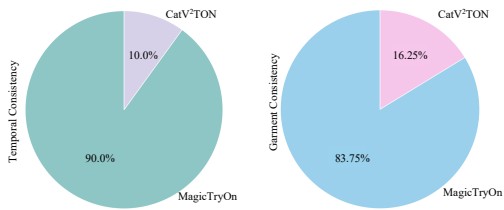

Figure 13: User study visualization. Participants were surveyed on spatiotemporal consistency and garment consistency.

Table 8: Comparison between the line estimation module and Canny edge extraction. Results are evaluated under identical testing conditions on the ViViD dataset. The best results are in red. $p$ and $u$ denote the paired and unpaired settings, respectively.

| Methods | VFID$_I^p\downarrow$ | VFID$_R^p\downarrow$ | SSIM↑ | LPIPS↓ | VFID$_I^u\downarrow$ | VFID$_R^u\downarrow$ |
|---|---|---|---|---|---|---|
| *w/* Canny edges | 8.9092 | 0.2772 | 0.8957 | 0.0609 | 15.0156 | 0.3469 |
| *w/* Line estimation | 8.4030 | 0.2346 | 0.9011 | 0.0602 | 14.7147 | 0.3200 |

in the Fig. 13. When asked to choose the video with better temporal consistency, considering smooth motion, absence of flickering artifacts, and visual stability, our method is selected 360 times out of 400 total responses (90%), significantly outperforming CatV$^2$TON (40 out of 400, 10%). In the garment consistency assessment, which measures the faithfulness of the generated garment to the target in terms of color, style, and structure, our method again receives a dominant preference with 335 out of 400 responses (83.75%), compared to CatV$^2$TON's 65 (16.25%). Participants showed a clear preference for MagicTryOn over CatV$^2$TON on both axes. These results substantiate that MagicTryOn delivers more stable and more faithful try-on videos, aligning with real-world user perception.

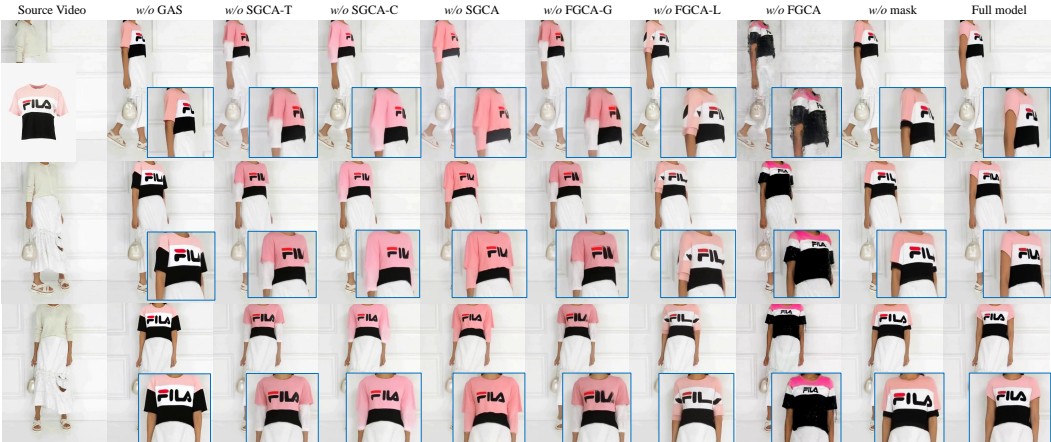

Figure 14: Visual comparison of the ablation variants. Please **zoom-in** for better visualization.

A.12    VISUAL RESULTS OF ABLATION STUDIES

Fig. 14 presents a visual comparison of the ablation variants. Removing GAS (garment-aware spatiotemporal RoPE) degrades cross-frame consistency and causes noticeable drift of garment features. When SGCA is partially removed, *w/o* SGCA-T (without text semantics) more often yields color and style deviations, while *w/o* SGCA-C (without CLIP semantics) leads to mismatches in garment category. Disabling SGCA entirely weakens global style and category constraints. The FGCA ablations show that *w/o* FGCA-G (without the appearance stream) produces blurred or faded high-frequency details such as logos and lettering, whereas *w/o* FGCA-L (without the structure stream) makes the silhouette and boundaries more prone to deformation and misalignment. Removing FGCA altogether simultaneously degrades texture and structure, often introducing blocky or smearing artifacts. Eliminating the mask-aware loss reduces optimization emphasis on garment regions, lowering regional consistency. In contrast, the Full model combines SGCA for semantic guidance, FGCA for appearance and structure feature guidance, GAS for spatiotemporal anchoring, and mask-aware reinforcement, maintaining silhouette and details stably across frames and achieving the best garment fidelity and spatiotemporal consistency.

A.13    LINE ESTIMATION VS. CANNY EDGE

We provide a comparison where the line estimation module is replaced with Canny edges. We evaluate both settings under the same testing conditions on the ViViD dataset, and the quantitative results are shown Tab. 8. As can be seen, when using the line maps extracted by Canny edges, the network performance drops slightly. We also provide visual comparisons between line maps extracted by line estimation module and those extracted by Canny edges in the Fig. 15. The line maps extracted by the line estimation module contain clearer contours and capture nearly all garment details, such as complex patterns, cuffs, collars, and other key elements that define garment structure. In contrast, the line maps produced by the Canny edges method fail to capture many of these important details. This explains why the generation performance drops when using the Canny line maps.

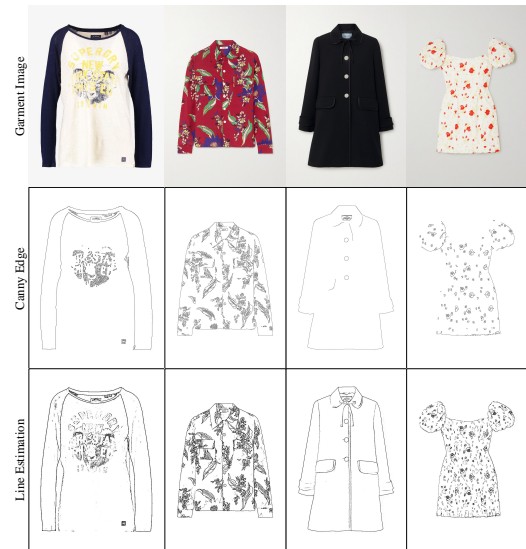

Figure 15: Visual comparison of line maps.

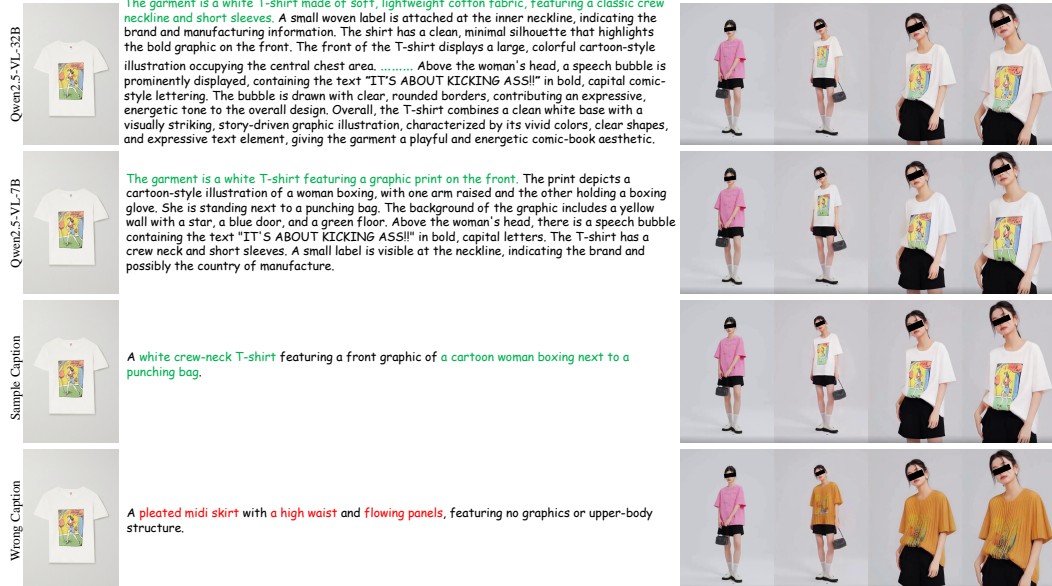

Figure 16: Comparison of generation results under four types of garment captions: 7B-generated, 32B-generated, simple-correct, and manually incorrect. Please **zoom-in** for better visualization.

## A.14 IMPACT OF CAPTION QUALITY

We perform an additional ablation study to investigate the influence of caption quality on the resulting try-on generation. we compare four different conditions, 1) using the caption generated by Qwen2.5-VL-7B, 2) using the caption generated by Qwen2.5-VL-32B, 3) using a very simple but semantically correct garment caption, and 4) using a manually written, completely incorrect garment caption (because the Qwen2.5-VL models rarely produce fully incorrect descriptions for garment images). The visual results are shown in Fig. 16. Although Qwen2.5-VL-32B produces more detailed garment descriptions, its generation quality is similar to that of Qwen2.5-VL-7B. A simple but correct garment caption can also produce strong try-on results. These findings show that as long as the caption provides a correct coarse description of the garment type and key features, high-quality try-on generation can be achieved. However, if the caption is completely incorrect

and does not match the garment category or main patterns of the target garment, the model cannot generate correct garment characteristics.

## A.15 UNDERPERFORMING SCENARIO

When the input garment mask is severely misaligned or semantically inconsistent, the model performance degrades. For example, during trouser try-on, if a mask corresponding to an upper-body garment is provided instead, the try-on performance underperforms. However, this issue is not specific to our method; it is a common limitation shared by all mask-based virtual try-on approaches.

