# OpenReview forum: "MagicTryOn: Harnessing Diffusion Transformer for Garment-Preserving Video Virtual Try-on"
_ICLR.cc/2026/Conference — ICLR 2026 Conference Withdrawn Submission_

### Official Review · Reviewer_qxVi · 2025-10-18

**Soundness:** 2
**Presentation:** 3
**Contribution:** 2
**Rating:** 6
**Confidence:** 4

**Summary:**

This paper introduces MagicTryOn, a framework for video virtual try-on (VVT) based on a diffusion transformer architecture. The primary goal is to enhance garment fidelity and spatiotemporal consistency in the generated videos. The authors propose three main contributions: 1) A fine-grained garment preservation strategy that decomposes the garment image into semantic, structural, and appearance cues, which are then injected into the denoising process via dedicated cross-attention mechanisms. 2) A garment-aware spatiotemporal rotary position embedding (GAS RoPE) designed to improve temporal stability by encoding relative positional information for garment tokens across frames. 3) The use of distribution-matching distillation to create a "Turbo" version of the model, enabling real-time inference in just four steps. The method is evaluated on standard VVT datasets like ViViD and VVT, where it reports state-of-the-art performance on several metrics.

**Strengths:**

*Strong Quantitative Results*: The paper presents impressive quantitative results on the ViViD and VVT datasets (Tables 1, 2, 5), consistently outperforming prior state-of-the-art methods across multiple metrics (VFID, SSIM, LPIPS). This suggests the proposed combination of techniques is highly effective.

*High-Speed Inference*: The development of MagicTryOn-Turbo, a distilled version capable of generating a 64-frame video in ~6.7 seconds, is a significant practical contribution. Achieving such a substantial speed-up (reportedly 50x) while maintaining strong performance makes the method much more viable for real-world applications.

*Well-Written Paper*: The paper is generally well-structured, clearly written, and easy to follow. The methodology is explained in sufficient detail, and the figures (especially Figure 2) are helpful in understanding the overall architecture.

**Weaknesses:**

*Lack of Video Evidence for Key Claims (Major Concern)*: The paper's primary claims revolve around improving spatiotemporal consistency and reducing temporal jitter. However, all comparisons—both against SOTA methods and in the ablation study—are presented as sequences of still frames. Still frames are insufficient for evaluating temporal dynamics and can be susceptible to cherry-picking. For a video generation task, providing actual video results in the supplementary material (e.g., via an anonymous repository) is crucial for validating claims of temporal superiority. The absence of video comparisons severely weakens the paper's persuasiveness.

*Overstated Novelty of Some Components*:
- Full Self-Attention: The paper mentions employing full self-attention to unify spatial and temporal modeling. However, this is not a novel contribution to the VVT domain, as it was previously utilized in CatV2TON. Moreover, the base model used, Wan2.1, already incorporates this architecture. The authors appear to be leveraging an existing feature of the backbone rather than introducing a new concept.
- Garment-Aware Spatiotemporal RoPE (GAS RoPE): The novelty of GAS RoPE is not clearly distinguished from prior work. The w/o GAS ablation simply removes RoPE for the prepended garment token, which would obviously degrade performance due to the lack of spatial alignment information. Methods like CatV2TON also treat the garment reference as a special token (akin to a first frame) to provide conditioning. A more detailed explanation of how GAS RoPE is fundamentally different and more effective is needed.
- Unfair Comparison with the Base Model: In Table 2, the authors compare MagicTryOn with a fine-tuned version of Wan2.1-I2V to demonstrate the effectiveness of their proposed modules. However, the paper states that the foundational model for MagicTryOn is Wan2.1-Fun-Control, which is an improved version of Wan2.1-I2V fine-tuned on additional data. This comparison is unfair. To isolate the true benefit of the proposed modules, the comparison should be against a fine-tuned Wan2.1-Fun-Control baseline, not the weaker Wan2.1-I2V.

*Questions Regarding the Ablation Study*:
- Magnitude of Impact: The ablation results in Table 3 are suspicious. Every single ablated component leads to a very significant drop in performance across nearly all metrics. It is unusual for each module to have such a large, independent impact. A more convincing approach would be an additive study, starting from a baseline and progressively adding each new component to show its marginal gain.
- Impact of Mask-Aware Loss: The performance drop from removing the mask-aware loss (w/o mask) is surprisingly large, nearly comparable to removing the entire Feature-Guided Cross-Attention (FGCA) module. Typically, such a loss term provides a modest refinement rather than being a cornerstone of performance. This counter-intuitive result requires a thorough explanation. Furthermore, mask-aware loss can risk information leakage from the mask shape, causing the model to generate garments that unnaturally conform to the mask boundaries. The paper does not discuss how this potential issue is mitigated.

*Potential Technical Inaccuracy*: Equation (5) formulates the training objective as a noise prediction loss (||εθ - ε||²). However, the Wan2.1 backbone model, on which this work is based, is known to use velocity prediction. This discrepancy should be clarified.

**Questions:**

*Component Choices*:
- Could you justify the use of a dedicated line estimation module (Pan, 2025) over a simpler, classic method like Canny edge detection? Was an ablation performed to compare different line extraction methods?
- For text descriptions, you used Qwen2.5-VL-7B. How sensitive is the model's performance to the quality and accuracy of these generated captions? Was a more powerful model, such as the 32B variant, considered to potentially provide more accurate descriptions?

*Interpretability*: To better demonstrate the contribution of the decomposed structural cues, could you provide visualizations of attention maps? This could offer a more direct and intuitive view of how the Line Tokens specifically contribute to the generation of sharp garment details, patterns, and logos.

*Ablation Study Clarifications*:
- Could you provide an explanation for the unexpectedly large performance drop when the mask-aware loss is removed?
- How did you prevent the mask-aware loss from causing the model to overfit to the mask shape, which can lead to unrealistic garment deformations, especially when the agnostic mask is not perfectly accurate? Video results for this specific ablation would be very informative.

*Video Results*: Will you be providing video comparisons to substantiate the claims of superior temporal consistency? (e.g., via an anonymous repository) This would include comparisons against SOTA methods and videos for the key ablation studies (e.g., with and without GAS RoPE, with and without mask-aware loss).

---

> ### Author Response · Authors · 2025-11-21
> **Rebuttal to Reviewer qxVi (Part 1)**
>
> *We sincerely thank you for taking the time to review our work and for your insightful comments. We hope the responses below adequately address your concerns. The updated content is highlighted in **Purple** in the revised version.*
>
> ### 1. Video Evidence (Major Concern)
>
> Thank you for your comment.
> **In the initial submission**, we provided video results of MagicTryOn in the supplementary material, **including Unconstrained Settings.mp4 and Multi-garment Scenarios.mp4**.
>
> **In the revised submission**, we add video comparisons with other methods (***Video Comparison with Other Methods.mp4***), video results of the ablation study (***Ablation Study Video.mp4***), and results under cross-category garment scenarios (***Cross-Category Garment Scenarios.mp4***) to the supplementary material.
>
> **We hope that these video results can address your main concerns and questions about the video results.**
>
> ### 2. Full Self-Attention
>
> Thank you for your comment.
> We do not claim full self-attention as a primary contribution of this work. The core innovations of our method focus on the following three aspects (Lines 78-101 in the manuscript):
>
> (1) We decompose garment information into semantic, structural, and appearance cues, and explicitly inject them into the diffusion denoising network. This multi-attribute garment conditioning differs from the designs in ViViD and CatV2TON, and serves as a task-specific contribution of MagicTryOn.
>
> (2) We extend rotary position encoding into a garment-aware spatiotemporal correspondence mechanism, enabling consistent anchoring of the same garment across frames. This design specifically addresses the challenge of preserving garment identity.
>
> (3) We introduce distribution-matching distillation, which allows the model to produce high-quality results within only four inference steps. This significantly improves practical usability and provides an efficient baseline for future research.
>
> **These contributions enable MagicTryOn to serve as a strong baseline that advances the development of video virtual try-on.**
>
> ### 3. GAS RoPE
>
> Thank you for your comment.
> The difference from prior work such as CatV2TON is that it treats the garment as an additional frame concatenated before entering the DiT blocks and encodes it using the standard RoPE. However, there is no explicit mechanism that enforces cross-frame correspondence. The standard RoPE in DiT models relative spatial positions and local relationships, without providing temporal alignment for the garment reference.
> **Our proposed GAS RoPE introduces garment-aware spatiotemporal correspondence by binding the garment tokens to consistent temporal anchors. This provides explicit structural alignment across frames, preserves garment identity throughout the sequence.**
>
> ### 4. Comparison with Wan2.1-Fun-Control
>
> Thank you for your comment.
> **In the revision, we add a comparison with Wan2.1-Fun-Control.** We fine-tune Wan2.1-Fun-Control using the same try-on dataset and experimental setup as our method. The quantitative results are shown below.
>
> | Methods     | $VFID^p_I$ ↓    | $VFID^p_R$ ↓ | SSIM ↑ | LPIPS ↓ | $VFID^u_I$ ↓  | $VFID^u_R$ ↓   |
> |-------------|--------|---------|--------|-----------|--------|---------|
> | Wan2.1-Fun-Control          | 14.2180    | 0.7113     | 0.8529 | 0.0818 | 19.9284    | 0.8656     |
> | MagicTryOn   | **8.4030** | **0.2346**  | **0.9011** | **0.0602**    | **14.7147** | **0.3200**  |
>
> We observe that MagicTryOn outperforms the fine-tuned Wan2.1-Fun-Control. **These results are updated in Table 2 of the revised version.**

---

> > ### Comment · Reviewer_qxVi · 2025-11-25
> > **Follow-up on the comparison with Wan2.1-Fun-Control**
> >
> > Thank you for conducting the additional experiment. However, I noticed an unexpected inconsistency in the reported results. The fine-tuned Wan2.1-Fun-Control is inferior to the performance of CatV2TON as reported in Table 1 of the main paper.
> > Theoretically, Wan2.1-Fun-Control is built upon a significantly more powerful foundation model compared to CatV2TON and natively supports reference image control (as documented in the official VideoX-Fun repository). One would reasonably expect a fine-tuned version of such a strong backbone to outperform the previous state-of-the-art. The fact that this strong baseline underperforms suggests it might not have been fully optimized during fine-tuning. Could the authors explain this discrepancy? This raises a concern about whether the baseline is strong enough to fairly evaluate the proposed method's contribution.

---

> > > ### Author Response · Authors · 2025-11-27
> > > **Response: comparison with Wan2.1-Fun-Control**
> > >
> > > We sincerely thank the reviewer for the follow-up comment. Regarding this concern, we would like to further explain as follows.
> > >
> > > First, the fine-tuning of Wan2.1-Fun-Control is conducted fully and rigorously. We follow the same training configuration as stated in the main paper (learning rate, video resolution, the total number of training iterations, etc.), and the training loss converges stably. Therefore, the performance does not stem from insufficient optimization.
> > >
> > > Second, CatV2TON and Wan2.1-Fun-Control differ fundamentally in task objectives and architectural design, so their performance cannot be compared by backbone size alone.
> > >
> > > 1. CatV2TON is a highly task-specialized video try-on model. Its architecture includes dedicated reference network, adaptive clip normalization, and an overlapping clip-based inference strategy, all of which provide strong inductive biases tailored to the try-on task.
> > >
> > > 2. Wan2.1-Fun-Control is a general video generation model. Although it supports reference image control, it does not incorporate any modules specifically designed for video try-on. As a result, even after fine-tuning, its performance on this specialized task does not reach the level of the task-optimized CatV2TON.
> > >
> > > **This further highlights the value of our proposed method. Our module is designed to effectively leverage the foundation model’s capacity for video try-on, enabling it to achieve strong task-specific performance.**

---

> ### Author Response · Authors · 2025-11-21
> **Rebuttal to Reviewer qxVi (Part 2)**
>
> ### 5. Additive Study
>
> Thank you for your suggestion.
> **In the revised version (Section 4.4), we supplement additive experiments that progressively incorporate each module into the bare model to more clearly demonstrate the marginal contribution of every component.**
> Specifically, starting from the Bare Model, we gradually add the SGCA module, the FGCA module, the mask loss, and the GAS RoPE.
> All variants use the same training environment described in Section 4.4 of the paper.
> The quantitative results on the ViViD test set with the resolution of 384×512 and 64 frames are as follows:
>
> | Methods     | $VFID^p_I$ ↓    | $VFID^p_R$ ↓ | SSIM ↑ | LPIPS ↓ | $VFID^u_I$ ↓  | $VFID^u_R$ ↓   |
> |-------------|--------|---------|--------|-----------|--------|---------|
> | Bare Model  |     21.9270    |   1.2376    | 0.8087    |   0.1181    | 28.2298      | 1.2003  |
> | + SGCA      |     19.2988    |  0.9313     | 0.8329    |   0.1007    | 25.3479      | 1.0360  |
> | + SGCA + FGCA   | 18.0794    | 0.7001      | 0.8630    |   0.0839    | 23.3136      | 0.8925  |
> | + SGCA + FGCA + mask loss   | 15.4081 | 0.5252  | 0.8704 |   0.0791  | 20.2338      | 0.7249  |
> | + SGCA + FGCA + mask loss + GAS   | **12.0640** | **0.2019**  | **0.8852** |  **0.0747**   | **18.0523** |  **0.5068** |
>
> **As shown, each incremental component consistently improves performance, with clear and steady gains across all metrics as the SGCA, FGCA, mask-aware loss, and GAS modules are progressively incorporated. We update the above results in Section 4.4 of the revised version.**
>
> ### 6. Impact of Mask-Aware Loss
>
> **(1) Regarding the impact of the mask-aware loss**
>
> The mask-aware loss serves as the supervisory signal that constrains the semantic, structural, and appearance garment cues to the correct garment region. In try-on videos, a large portion of pixels belongs to the human body and background, while the actual garment region occupies only a small proportion. Removing this supervision causes the model to lose explicit guidance on the garment area during training, making the optimization of SGCA, FGCA, and GAS RoPE unstable and diluted, ultimately leading to a noticeable performance drop.
>
> **(2) Regarding the issue of mask shape**
>
> First, the masks used during training do not closely follow the true garment boundaries; instead, they are coarse masks obtained through dilation, whose coverage intentionally includes a small portion of the human body and background. This prevents the model from overfitting to the exact contour of the mask.
>
> Second, since a small portion of the masks in the training set are imprecise, the model is exposed to coarse region cues of varying quality and shapes during training. As a result, it is encouraged to rely on semantic, structural, and appearance cues to identify the garment region rather than simply following the mask boundary. This process actually enhances the model’s robustness to mask noise.
>
> **We provide results under cross-category garment scenarios (Cross-Category Garment Scenarios.mp4) in the supplementary material to demonstrate that the garment does not deform according to the shape of the mask**.
> **We hope these responses address your concerns regarding the mask and your questions about the ablation study clarifications.**
>
> ### 7. Potential Technical Inaccuracy
>
> Thank you for your careful correction. Regarding Equation (5), we use the same velocity prediction as Wan2.1. The definition of the symbols is missing in Equation (5), and **we have corrected this issue in the revised version**.

---

> ### Author Response · Authors · 2025-11-21
> **Rebuttal to Reviewer qxVi (Part 3)**
>
> ### 8. Questions about Component Choices
>
> *Q1. Could you justify the use of a dedicated line estimation module (Pan, 2025) over a simpler, classic method like Canny edge detection? Was an ablation performed to compare different line extraction methods?*
>
> Canny edge detection is insufficient for extracting garment structures. The edges obtained by Canny cannot depict the complete garment silhouette, and for garments with complex patterns, the extracted edges are often fragmented and discontinuous. In contrast, the line estimation module is specifically designed for estimating structural lines. It can stably and fully extract the garment contour and also produces high-quality line representations for garments with complex patterns.
>
>
> *Q2. For text descriptions, you used Qwen2.5-VL-7B. How sensitive is the model's performance to the quality and accuracy of these generated captions? Was a more powerful model, such as the 32B variant, considered to potentially provide more accurate descriptions?*
>
> Regarding the model’s sensitivity to caption quality, it only requires the key garment attributes, such as the garment category, color, main pattern, and overall style at a coarse level. Qwen2.5-VL-7B can reliably and accurately describe garment attributes such as category, color, pattern, and style. Using a larger Qwen2.5 model to generate garment captions does not provide significant performance improvements.
>
>
> ### 9. Questions about Interpretability
>
> *Q3. To better demonstrate the contribution of the decomposed structural cues, could you provide visualizations of attention maps? This could offer a more direct and intuitive view of how the Line Tokens specifically contribute to the generation of sharp garment details, patterns, and logos.*
>
> **In the revised version, we provide visualizations of the Line-Token Attention Map, as shown in Figure 6.** We observe that the attention weights are primarily concentrated on key garment regions, such as sharp details, patterns, and logos. This helps the model better understand fine-grained structural cues of garments and produce more accurate try-on results.

---

> > ### Comment · Reviewer_qxVi · 2025-11-25
> > **Follow-up on Component Choices (Line Estimation & Captioning)**
> >
> > Regarding Q1 (Line Estimation vs. Canny):
> >
> > While I acknowledge that the specific line estimation module (Pan, 2025) produces visually cleaner contours than Canny, I remain unconvinced that this difference significantly impacts the final generation quality without empirical evidence. Modern diffusion models are known for their robustness and ability to generate high-quality results even with noisy or sparse conditions (e.g., ControlNet often works well with standard Canny maps).
> > Since the "visual quality" of the condition map does not necessarily equate to the "generation quality" of the video, could the authors provide a comparison where the line estimation module is replaced by Canny edges? If the final try-on results are similar, the inclusion of a specialized line estimator would be an unnecessary complexity.
> >
> > Regarding Q2 (Sensitivity to Caption Quality):
> >
> > The authors claim that Qwen2.5-VL-7B is "sufficient" for coarse attributes. However, since the proposed SGCA (Semantic-Guided Cross-Attention) module is a key contribution of this paper, the reliability of its input is critical. VLMs, even capable ones, suffer from hallucinations and can misclassify simple attributes like garment category or patterns, which would directly mislead the SGCA module.
> > I suggest two clarifications:
> > - Accuracy Assessment: Have you performed a manual check or a cross-check (using a stronger model) on a subset of the data to verify the accuracy of the 7B model's captions (especially for garment categories and main patterns)?
> > - Impact of Error: How does the model behave when the caption is incorrect? Since you argue the model only needs "coarse" levels, does the generation fail or degrade if the VLM provides a wrong category?

---

> > > ### Author Response · Authors · 2025-11-27
> > > **Response: Component Choices (Line Estimation & Captioning)**
> > >
> > > ### Line Estimation vs. Canny
> > >
> > > Thank you for your insightful comment. We provide a comparison where the line estimation module is replaced with Canny edges. We evaluate both settings under the same testing conditions on the ViViD dataset, and the quantitative results are shown below.
> > >
> > > | Methods     | $VFID^p_I$ ↓    | $VFID^p_R$ ↓ | SSIM ↑ | LPIPS ↓ | $VFID^u_I$ ↓  | $VFID^u_R$ ↓   |
> > > |-------------|--------|---------|--------|-----------|--------|---------|
> > > | *w/* Canny edges          |  8.9092   |   0.2772   | 0.8957 | 0.0609 |  15.0156   |  0.3469    |
> > > | *w/* Line estimation   | **8.4030** | **0.2346**  | **0.9011** | **0.0602**    | **14.7147** | **0.3200**  |
> > >
> > > As can be seen, when using the line maps extracted by Canny edges, the network performance drops slightly. We also include visual comparisons between line maps extracted by line estimation module and those extracted by Canny edges in the revised version. **Please refer to Figure 15 in the revised version.**
> > >
> > > **The line maps extracted by the line estimation module contain clearer contours and capture nearly all garment details, such as complex patterns, cuffs, collars, and other key elements that define garment structure**. In contrast, the line maps produced by the Canny edges method fail to capture many of these important details. This explains why the generation performance drops when using the Canny-based line maps.
> > >
> > > **We add the above results in Section A.13 of the revised version.**
> > >
> > > ### Sensitivity to Caption Quality
> > >
> > > For the quality of captions generated by the Qwen2.5-VL-7B model, we performed a manual check when constructing the garment captions. The check focused on verifying consistency in garment category and main appearance descriptions. From the subset we inspected, the generated captions match the true garment category, main patterns, and key appearance elements (such as prints and text).
> > >
> > > Regarding the model behaves when the caption is incorrect, we conduct an additional analysis. For garment image caption, current vision-language models are very reliable. Both Qwen2.5-VL-7B and the larger Qwen2.5-VL-32B can consistently produce descriptions that match the garment category and main textures.
> > >
> > > To evaluate how garment captions influence the generation results, we compare four different conditions:
> > > 1. using the caption generated by Qwen2.5-VL-32B,
> > > 2. using the caption generated by Qwen2.5-VL-7B,
> > > 3. using a very simple but semantically correct garment caption,
> > > 4. using a manually written, completely incorrect garment caption (because the Qwen2.5-VL models rarely produce fully incorrect descriptions for garment images).
> > >
> > > **Please refer to Section A.14 (Figure 16) in the revised version.**
> > > Although Qwen2.5-VL-32B produces more detailed garment descriptions, its generation quality is similar to that of Qwen2.5-VL-7B. A simple but correct garment caption can also produce strong try-on results. These findings show that as long as the caption provides a correct coarse description of the garment type and key features, high-quality try-on generation can be achieved. However, if the caption is completely incorrect and does not match the garment category or main patterns of the target garment, the model cannot generate correct garment characteristics.

---

### Official Review · Reviewer_xRnT · 2025-10-22

**Soundness:** 2
**Presentation:** 2
**Contribution:** 2
**Rating:** 4
**Confidence:** 5

**Summary:**

The paper introduces MagicTryOn, a diffusion transformer-based framework for garment-preserving video virtual try-on (VVT) that aims to improve both garment-detail fidelity and spatiotemporal consistency in generated try-on videos. The approach leverages a fine-grained garment-preservation strategy (decomposing cues into semantic, structure, and appearance), introduces a garment-aware spatiotemporal rotary position embedding (RoPE) for enhanced temporal consistency, and utilizes a mask-aware loss to enforce garment-region fidelity. Furthermore, it integrates a distribution-matching distillation to accelerate inference without compromising quality. The authors present quantitative and qualitative improvements over several state-of-the-art methods on public VVT benchmarks, supported by ablations and user studies.

**Strengths:**

1. Innovative Spatiotemporal Encoding: The extension of RoPE to “garment-aware spatiotemporal RoPE” is principled and directly addresses temporal instability; the subsuming of garment tokens into the full self-attention with grid extension is mathematically described and justified.
2. Thorough Ablations: Table 3 and Figure 12 provide a detailed ablation of key architectural modules, with analyses that specify the performance and qualitative impact of removing each token stream or loss component, which aids scientific transparency.
3. Clear writting.

**Weaknesses:**

1. Limited Novelty in Architectural Choices: While the garments’ semantic/structural/appearance decomposition and the cross-attention wiring are interesting, the overall method mainly combines existing mechanisms from prior VVT and diffusion transformer literature, such as patchification, CLIP feature usage, full self-attention, and cross-token fusion. As evident from the Related Work and as per foundational methods like ViViD, CatV2TON, or Hunyuan-DiT (all cited in the main text), many core strategies (semantic cross-attention, diffusion transformer backbone, pose-agnostic inputs, multi-modal fusion) are also used in recent related architectures.
2. In addition, the semantic/structural/appearance sequence tokens are redundant and can be replaced by a limited number of keyframes, thereby further enhancing efficiency and reducing preprocessing requirements.
3. Based on the video materials provided, the clothing replacements demonstrated in the paper appear limited—for instance, only showcasing exchanges between different T-shirts or between various pairs of jeans, while failing to present cross-category swaps such as skirts versus trousers. This limitation may be attributed to constraints inherent in the mask-based approach used.

**Questions:**

See weakness.

---

> ### Author Response · Authors · 2025-11-21
> **Rebuttal to Reviewer xRnT**
>
> *We sincerely thank you for taking the time to review our work and for your insightful comments. We hope the responses below adequately address your concerns. The updated content is highlighted in **Orange** in the revised version.*
>
> ### 1. Architectural Choices
>
> Thank you for your comment. **The contribution of MagicTryOn does not lie in redeveloping or choosing to use an architecture that is entirely different from existing systems.** Instead, **it focuses on addressing the key challenges of video virtual try-on by introducing several modules that lead to substantial performance improvements.**
>
> For clarity, we summarize the main innovations of our method as follows:
>
> (1) We decompose garment information into three attributes: semantic, structural, and appearance cues, and explicitly inject them into the denoising network. **This design is not a standard practice in existing methods such as ViViD and CatV2TON.**
>
> (2) We extend rotary position encoding to a garment-aware spatiotemporal RoPE, providing explicit cross-frame correspondence constraints that enable the model to preserve garment identity even under human motion.
>
> (3) We adopt distribution-matching distillation to compress the inference process to four diffusion steps while maintaining try-on quality, achieving a 50$\times$ speed-up. **The accelerated version of MagicTryOn not only improves practical usability but also enables our method to serve as an efficient baseline for future research.**
>
> Overall, the contributions of MagicTryOn lie in three aspects: garment-conditioned decomposition modeling, spatiotemporal consistency constraints, and distillation-based acceleration. **These key designs go far beyond a standard combination of existing architectures. Moreover, the experimental results validate the significant performance gains brought by these designs, establishing MagicTryOn as an *important baseline* in the virtual try-on community and advancing the development of video virtual try-on.**
>
> ### 2. Keyframes
>
> Thank you for your suggestion. Regarding the question of whether the semantic, structural, and appearance tokens are redundant and can be replaced by a few key frames, we would like to clarify two points:
>
> (1) **These tokens are not redundant but complementary, and each serves a distinct informational role.** The semantic tokens provide category, color, and part-level garment information. The structural tokens describe garment geometry and contours. The appearance tokens preserve fine-grained texture details. **A few key frames cannot simultaneously capture all three types of garment information across these different dimensions.** Moreover, the ablation study further confirms the effectiveness of these tokens.
>
> (2) Although we adopt three types of tokens, **the overall efficiency of the model remains very high**, as we compress the inference process to four steps through distribution-matching distillation, achieving a 50 $\times$ acceleration.
>
> In summary, these tokens serve as complementary and non-interchangeable conditional representations that significantly enhance semantic control, structural preservation, and appearance reconstruction. Moreover, even with all three types of tokens, MagicTryOn still maintains very high inference efficiency.
>
> ### 3. Cross-Category Garment Scenarios
>
> **We provide several examples of cross-category cases in the initial submission.** Figure 4 shows a transformation from a skirt to trousers. Figure 7 presents a case from a short-sleeve T-shirt to a long-sleeve shirt, and Figure 11 shows another example from short sleeves to long sleeves.
>
> **In the revised version, we include additional cross-garment examples in Figure 5**, such as trousers to shorts, trousers to skirts, and skirts to trousers. We also provide **cross-category try-on video results** in the supplementary material of the revised version. Please refer to ***Cross-Category Garment Scenarios.mp4***. As shown in Figure 5 of the revised manuscript, and with many more results provided in the supplementary material, MagicTryOn can generate garment contours and structures that match the target clothing, without being restricted by the mask shape of the original garment.
>
> In addition, mask-based methods do not impose restrictions on cross-category garment transformation. A mask that roughly corresponds to the target garment region is sufficient to enable any garment category to be transferred.

---

> > ### Comment · Reviewer_xRnT · 2025-11-24
> >
> > Thanks for authors' response.
> > 1.  The injection of attributes semantic, structural, and appearance cues have been widely used in image generation and editing, such as Anydoor (edge), Animateanyone (pose). This paper use these existing techniques in DiT backbone, while it is not novel enough.
> > 2. The distillation is efficient while it is more like an engineer improvement since the distillation method is following DMD.
> > 3. The mask based pipeline decreases the freedom of generation because of the leak of garment shape in the self-supervised learning. So I guess the cross-category garment transfer is not robustness in the proposed framework.

---

> > > ### Author Response · Authors · 2025-11-24
> > > **Follow-up Rebuttal to Reviewer xRnT**
> > >
> > > We sincerely thank the reviewer for the follow-up comment. Our responses to the three comments are summarized below.
> > >
> > > ### Comment 1:
> > > We would like to clarify the main differences between our work and existing methods.
> > >
> > > First, AnyDoor (edge) is an image editing method and does not consider cross-frame consistency or video sequences. AnimateAnyone (pose) is an image-to-video method, but it mainly animates a single portrait rather than performing video-based garment transfer. **In contrast, video try-on is much more challenging**: the model must keep garment identity, shape, and details consistent as the person moves through the video.
> > >
> > > Second, **existing methods usually use only one type of cue** (for example, edges in AnyDoor or poses in AnimateAnyone). In our work, **we are the first to separate garment information into three cues (semantic, structural, and appearance) and use them together in a unified diffusion framework**. To the best of our knowledge, such a disentangled and unified design has not been explored in video try-on before.
> > >
> > > In addition, **our injection strategy is also different**. Instead of simple concatenation (as in AnimateAnyone) or using a reference network (as in AnyDoor), we use disentangled cross-attention to inject the three garment cues separately. This helps the cues stay consistent across the whole video and improves the overall quality.
> > >
> > > **Therefore, our contribution is not simply to use these existing techniques. Compared with image generation/editing methods and existing video try-on approaches, our method provides clear and substantial novelty.**
> > >
> > > ### Comment 2:
> > > Regarding the comment that our method looks more like an engineering improvement, we would like to provide some clarification.
> > >
> > > Although our acceleration strategy is built on top of DMD, **it is not a simple engineering copy of existing frameworks. Instead, we introduce new technical extensions and task-specific designs for the controlled Video-to-Video (V2V) setting**. Current DMD-based acceleration works mainly focus on Text-to-Video (T2V) tasks, such as CausVid(CVPR2025) and Self-Forcing (NeurIPS2025). **In comparison, controlled V2V video try-on is more complex**: the model must handle the person, garment, pose control, and cross-frame consistency at the same time, making existing DMD methods not directly applicable.
> > >
> > > Our technical novelty mainly lies in two aspects:
> > >
> > > (1) Existing DMD acceleration methods rely on a standard T2V causal DiT, while **we build the first causal DiT designed specifically for controlled V2V**, supporting multiple control signals and adapting to the complex constraints of video try-on.
> > >
> > > (2) **On top of the KV-cache, we further introduce three garment-related caches (CLIP image cache, garment cache, and line cache)**. These caches not only speed up inference but also help maintain garment identity and details, which prior T2V acceleration methods do not handle.
> > >
> > > Based on these contributions, **our method is not just an engineering optimization. Instead, it provides the first systematic extension of the DMD framework to controlled video generation**. To the best of our knowledge, **MagicTryon-Turbo is the first distilled system that accelerates controlled V2V video generation, offering a new technical perspective for future research on efficient controlled video synthesis**.
> > >
> > > ### Comment 3:
> > > Regarding the mask question, we would like to clarify the following:
> > >
> > > First, in both image-based and video-based virtual try-on, most existing methods use masks as a basic constraint to locate the garment region. **Using masks to separate the garment from the person is a widely adopted and well-validated strategy in the community**, so our design follows established practice.
> > >
> > > Second, our mask pipeline is a weak guidance rather than a strong constraint. **These masks do not encode the actual garment shape; they only provide rough spatial regions**. The final garment shape in the generated video is fully **determined by the reference garment cues**, not by the mask itself.
> > >
> > > In addition, we compare MagicTryOn with other methods on cross-category garment transfer. We evaluate 660 video cases where the source and target garments belong to different categories. The results are shown below:
> > >
> > > | Method              | CatV2TON  | MagicTryOn |
> > > |---------------------|----------|-----------------|
> > > | $VFID^u_I$ ↓        |  31.8946 |     **17.6319**     |
> > > | $VFID^u_R$ ↓        |  1.5568 |     **0.8119**      |
> > >
> > > **Both our quantitative results and video examples show that our cross-category performance clearly surpasses existing open-source video try-on methods.**
> > > While masks may show limitations in a few rare and extreme cases, this is a common characteristic shared by all mask-based approaches rather than an issue specific to our framework. Importantly, these rare cases do not affect the overall robustness of our cross-category results, as supported by both our quantitative evaluation and visual examples.

---

> > > > ### Comment · Reviewer_xRnT · 2025-11-24
> > > >
> > > > Thanks for author's response.
> > > >
> > > > I.  believe that the task of video try-on is merely an application of video editing. Although the authors claim that certain designs are appearing for the first time in video try-on, this is essentially a form of borrowing, and it is difficult to consider it a significant contribution.
> > > >
> > > > 2.  The 50x acceleration in video try-on is indeed a highly practical achievement. Although it is an extension based on DMD, I believe this part should be highlighted in the main text rather than placed in the appendix.
> > > >
> > > > 3. I am aware that mask-based editing is commonly used in the field of image and video try-on, but I do not think this approach is still necessary today. For instance, methods like AnyDressing might better align with the practical needs of virtual try-on. In this work, inserting too many tokens into the DiT network significantly increases the computational cost.

---

> > > > > ### Author Response · Authors · 2025-11-24
> > > > > **Follow-up Rebuttal to Reviewer xRnT**
> > > > >
> > > > > We sincerely thank the reviewer for the follow-up comment. We would like to offer some further discussion and clarification.
> > > > >
> > > > > ### Comment 1:
> > > > > We would like to further clarify our contribution.
> > > > > Our method is not a simple reuse of existing techniques. Instead, **we introduce task-specific designs for the video try-on setting**. In deep learning, many works (such as those using attention or ControlNet) build on existing modules, but the key questions are: **whether the method is adapted to a new task**, **whether it includes new designs for that task**, and **whether it solves problems that previous methods could not handle**. In our method, the disentangled injection of three garment cues, the mechanisms for keeping garment identity, and the acceleration distillation, **are all designs created specifically for video try-on**. Our experiments show that these task-specific improvements overcome the limitations of previous methods and achieve very good results.
> > > > >
> > > > > ### Comment 2:
> > > > > Thank you for your positive feedback on our distillation acceleration. We appreciate your suggestion and will highlight this contribution in the revised main text.
> > > > >
> > > > > ### Comment 3:
> > > > > **The mentioned AnyDressing method is different from the standard video try-on task**. In typical video try-on, the goal is to replace the garment on a given person, while keeping the person’s identity, body shape, and appearance consistent across the whole video. **In contrast**, AnyDressing is essentially an image-based subject generation method. It applies garments driven by prompts and extra conditions (such as pose or face features). Its goal is not to replace the clothing on the original person, nor to keep the person’s body shape, height, or appearance. Therefore, it cannot directly meet the requirements of standard video try-on, which must keep the person and the temporal stability.
> > > > >
> > > > > **To reliably replace clothing on a given person, video try-on methods usually need a mask to mark the garment area.** This is a common and effective design in both image and video try-on tasks. In fact, AnyDressing also uses masks (see Eq. 9), which shows that spatial guidance from masks is still useful even in image-based methods.
> > > > >
> > > > > **Regarding the concern about computational cost**, MagicTryon-Turbo runs efficiently in practice and requires only 21.32 GB of GPU memory. We also compare the FLOPs, as shown below:
> > > > >
> > > > > | Method              | CatV2TON  | MagicTryOn-Turbo |
> > > > > |---------------------|----------|-----------------|
> > > > > | GPU memory       |  27.66G |    **21.32G**     |
> > > > > | Inference time        |  209.127s |     **6.69s**      |
> > > > > | FLOPs       |  3.61T |     **2.27T**      |
> > > > >
> > > > > As the results show, MagicTryon-Turbo has clear advantages in GPU memory, inference time, and FLOPs. This indicates that the extra tokens do not become a bottleneck, and our design is practical and efficient in real use.

---

### Official Review · Reviewer_HuPr · 2025-10-27

**Soundness:** 2
**Presentation:** 2
**Contribution:** 2
**Rating:** 4
**Confidence:** 3

**Summary:**

This paper introduces **MagicTryOn**, a diffusion transformer-based framework for **Video Virtual Try-On (VVT)** to address issues with inadequate garment fidelity and limited spatiotemporal consistency in existing methods.

The framework improves fine-detail fidelity by proposing a **fine-grained garment-preservation strategy** that disentangles and injects decomposed garment cues into the denoising process.

To enhance temporal consistency, the authors introduce a **garment-aware spatiotemporal rotary positional embedding (RoPE)** and further employ a distribution-matching distillation technique to enable real-time, four-step inference, ultimately demonstrating superior garment-detail fidelity and temporal stability over state-of-the-art methods.

**Strengths:**

1. The paper proposes a well-structured diffusion transformer framework tailored for video virtual try-on with clear motivation and technical contributions.
2. It introduces innovative modules, including fine-grained garment-preservation and garment-aware spatiotemporal RoPE, effectively enhancing detail fidelity and temporal consistency.
3. The method achieves real-time inference through distribution-matching distillation while maintaining strong performance, supported by comprehensive experiments.

**Weaknesses:**

1. The proposed design appears rather standard, primarily relying on a combination of strong pretrained encoders and DiT blocks with cross-attention. The architectural novelty and unique algorithmic contribution seem limited.
2. The framework integrates numerous large components—VAE, T5 encoder, CLIP encoder, Qwen-7B, and Wan2.1—resulting in a highly complex system. It remains unclear which specific modules in Figure 2 are initialized with Wan2.1 pretrained weights, as mentioned in line 315.
3. Given the scale of the model and the processing of 64 frames over 6.69 seconds, the claim of real-time inference seems questionable. Details regarding the evaluation setup, baseline comparisons, number of runs, and deviation bars are missing, making the performance claim less convincing.
4. The analysis section feels overloaded with too many individual use cases (nine in total), which makes it difficult to assess the true impact of each component. A more focused and compact ablation study combining related modules would make the results clearer and more insightful.

**Questions:**

See weaknesses

---

> ### Author Response · Authors · 2025-11-21
> **Rebuttal to Reviewer HuPr**
>
> *We sincerely thank you for taking the time to review our work and for your insightful comments. We hope the responses below adequately address your concerns. The updated content is highlighted in **Green** in the revised version.*
>
> ### 1. Architectural Novelty
>
> Thank you for your comment. **The contribution of this work does not lie in designing an entirely new architecture.** Instead, **it focuses on addressing the key challenges of video virtual try-on by introducing several substantive module designs that bring significant performance improvements.** We summarize the main innovations of our method as follows:
>
> (1) We decompose garment information into three attributes: semantic, structural, and appearance cues, and explicitly inject them into the denoising network. **This design is not a standard practice in existing methods such as ViViD and CatV2TON.**
>
> (2) We extend rotary position encoding to a garment-aware spatiotemporal RoPE, providing explicit cross-frame correspondence constraints that enable the model to preserve garment identity even under human motion.
>
> (3) We adopt distribution-matching distillation to compress the inference process to four diffusion steps while maintaining try-on quality, achieving a 50 $\times$ speed-up. **The accelerated version of MagicTryOn not only improves practical usability but also enables our method to serve as an efficient baseline for future research**.
>
> The experimental results validate the significant performance gains introduced by these designs, making **MagicTryOn an important baseline in the virtual try-on community and advancing the development of video virtual try-on**.
>
> ### 2. Specific Modules
>
> The Wan Encoder, Wan Decoder, VAE, and DiT Blocks in Figure 2 are initialized with the pretrained weights of Wan2.1. Among them, the Wan Encoder, Wan Decoder, and VAE are frozen.
> The T5 encoder is taken from google/umt5-xxl.
> The CLIP encoder is taken from OpenAI.
> Qwen-7B is taken from Qwen.
>
> ### 3. Details of the Inference Time
>
> Lines 376-377 and 402-403 of the paper already provide detailed descriptions of our evaluation setup and the baseline comparison protocol. The inference time reported in the paper corresponds to the runtime of a single execution. **To further strengthen the reliability of the conclusions, we additionally conduct 180 runs and report the average inference time and deviation range, as shown below.**
>
> | Methods     | Inference time Avg    | Inference time Std |
> |-------------|--------|---------|
> | ViViD             | 204.8639 | 0.1959  |
> | CatV2TON          | 209.1763 |  0.2045 |
> | MagicTryOn-Turbo  | 6.6879 | 0.0042  |
>
> These statistics are obtained under the same hardware configuration (H20 GPU), resolution settings, and 64-frame inference conditions as reported in the paper. **As shown, MagicTryOn-Turbo is not only significantly faster than existing methods but also exhibits extremely low standard deviation in runtime, indicating a highly stable inference process.**
>
> ### 4. Analysis Section
>
> **Our ablation study is designed to clearly isolate the independent contribution of each key component.** Therefore, we separately analyze the semantic, structural, and appearance cues, the GAS RoPE, the mask-aware loss in MagicTryOn.
>
> **To provide a more focused and compact ablation study by combining related modules, we design an additional set of ablation experiments.** Specifically,
> we begin with the Bare Model and gradually add the SGCA module, the FGCA module, the mask loss, and the GAS RoPE.
>
> All variants use the same training environment as described in Section 4.4 of the main paper. The quantitative results on the ViViD test set with a resolution of 384×512 and 64 frames are as follows:
>
> | Methods     | $VFID^p_I$ ↓    | $VFID^p_R$ ↓ | SSIM ↑ | LPIPS ↓ | $VFID^u_I$ ↓  | $VFID^u_R$ ↓   |
> |-------------|--------|---------|--------|-----------|--------|---------|
> | Bare Model  |     21.9270    |   1.2376    | 0.8087    |   0.1181    | 28.2298      | 1.2003  |
> | + SGCA      |     19.2988    |  0.9313     | 0.8329    |   0.1007    | 25.3479      | 1.0360  |
> | + SGCA + FGCA   | 18.0794    | 0.7001      | 0.8630    |   0.0839    | 23.3136      | 0.8925  |
> | + SGCA + FGCA + mask loss   | 15.4081 | 0.5252  | 0.8704 |   0.0791  | 20.2338      | 0.7249  |
> | + SGCA + FGCA + mask loss + GAS   | **12.0640** | **0.2019**  | **0.8852** |  **0.0747**   | **18.0523** |  **0.5068** |
>
> **As shown, each incremental component consistently improves performance, with clear and steady gains across all metrics as the SGCA, FGCA, mask-aware loss, and GAS modules are progressively incorporated.** We update the above results in **Section 4.4 of the revised version**.

---

### Official Review · Reviewer_XFjK · 2025-10-30

**Soundness:** 3
**Presentation:** 3
**Contribution:** 3
**Rating:** 4
**Confidence:** 5

**Summary:**

This paper proposes MagicTryOn, a DiT-based framework for video virtual try-on. It decomposes garments into semantic/structure/appearance cues, injects them via two cross-attention modules, and extends RoPE to garment-aware spatiotemporal positional encoding. The proposed method is evaluated on the ViViD dataset.

**Strengths:**

1. Experiments are conducted on both image-based and video-based datasets.
2. Ablation studies are conducted to evaluate the effectiveness of each component.

**Weaknesses:**

1. The claim of correctly maintaining “compositional relationships” in multi-garment try-on is only supported by qualitative Fig.4, with no quantitative metrics (e.g., VFID-I3D, SSIM for multi-garment sequences) or statistical analysis. This leaves the performance of multi-garment handling unsubstantiated.
2. The main comparison (Table 1) excludes recent state-of-the-art methods like DreamVVT (Zuo et al., 2025) or SwiftTry (Nguyen et al., 2025), which also focus on temporal consistency. This incomplete benchmarking makes it hard to contextualize MagicTryOn’s true standing in the current VVT landscape.
3. Some qualitative examples are not promising. For instance, in Figure 4, the shorts in the generated video underwent deformation to align with the mask shape of the skirt from the original video.
4. The paper lacks analysis of scenarios where the method may underperform.

**Questions:**

See above

---

> ### Author Response · Authors · 2025-11-21
> **Rebuttal to Reviewer XFjK**
>
> *We sincerely thank you for taking the time to review our work and for your insightful comments. We hope the responses below adequately address your concerns. The updated content is highlighted in **Magenta** in the revised version.*
>
> ### 1. Quantitative Metrics under Multi-garment Scenarios
>
> We conduct a quantitative comparison with CatV2TON in the multi-garment scenario. Since no paired data are available in this setting, we compute the $VFID^u_I$ and $VFID^u_R$ metrics:
>
> | Method              | CatV2TON  | MagicTryOn |
> |---------------------|----------|-----------------|
> | $VFID^u_I$ ↓        | 61.6164  | **26.1804**         |
> | $VFID^u_R$ ↓        | 14.1268  | **5.6258**          |
>
> Combining the quantitative metrics and visual comparisons, MagicTryOn outperforms existing methods in the multi-garment scenario. **We update the above quantitative comparison in Table 5 of the revised version**.
>
> ### 2. Comparison with DreamVVT and SwiftTry
>
> We compare our method with DreamVVT on the ViViD dataset. Since the code of DreamVVT is not publicly available, we perform a quantitative comparison, and the metrics are taken from the results reported in the original paper. The results are as follows:
>
> | Methods     | $VFID^p_I$ ↓    | $VFID^p_R$ ↓ | SSIM ↑ | LPIPS ↓ | $VFID^u_I$ ↓  | $VFID^u_R$ ↓   |
> |-------------|--------|---------|--------|-----------|--------|---------|
> | DreamVVT          | 11.0180| 0.2549  | 0.8737 | 0.0619    | 16.9468 | 0.4285  |
> | MagicTryOn   | **8.4030** | **0.2346**  | **0.9011** | **0.0602**    | **14.7147** | **0.3200**  |
>
> We compare our method with SwiftTry on the VVT dataset, and the quantitative results are as follows:
>
> | Methods     |  SSIM ↑ | LPIPS ↓ | $VFID^u_I$ ↓    | $VFID^u_R$ ↓ |
> |-------------|--------|---------|--------|-----------|
> | SwiftTry            | 0.887 | 0.066    | 3.589| 0.5340 |
> | MagicTryOn     | **0.917** | **0.024**    | **1.662** | **0.0053** |
>
> **We observe that MagicTryOn outperforms both DreamVVT and SwiftTry across all metrics. We update the above quantitative comparisons in Table 1 and Table 7 of the revised version**.
>
> ### 3. Qualitative Examples
>
> Thank you for your insightful comment. **We would like to clarify that the deformation of the shorts in Figure 4 is not caused by the mask shape of the skirt**. Instead, it results from natural motion-induced dynamics learned by the model, such as garment swinging and temporal deformation. In the revised version, we provide additional video frames in Figure 4 to further illustrate this behavior.
>
> **Video try-on in multi-garment scenarios is more challenging**. As shown in Figure 4, existing methods (including ViViD and CatV2TON) fail noticeably under this setting, whereas MagicTryOn maintains consistent appearance and detailed fidelity.
>
> In video try-on, the mask only serves as a binary indicator of the region to be regenerated; it does not constrain the model to shape the clothing according to the mask. **To further verify this, we include additional cross-garment try-on examples in the revised version.** **As shown in Figure 5 of the revised manuscript, and with many more results provided in the supplementary material**, MagicTryOn can generate garment contours and structures that match the target clothing, without being restricted by the mask shape of the original garment.
>
> ### 4. Underperforming Scenario
>
> Thank you for your suggestion. In the revised version (Section A.13), we add a discussion of the scenarios where the methods underperform, as follows.
>
> When the input garment mask is severely misaligned or semantically inconsistent, the model performance degrades. For example, during trouser try-on, if a mask corresponding to an upper-body garment is provided instead, the try-on performance underperforms. However, this issue is not specific to our method; it is a common limitation shared by all mask-based virtual try-on approaches.

---

### Author Response · Authors · 2025-12-01
**Brief Summary**

We sincerely thank the AC and all reviewers for their time and constructive feedback. Below we provide a brief summary based on the reviewers’ comments as well as our rebuttal and subsequent discussions.

### Strengths Highlighted by Reviewers
Reviewers highlighted the thorough ablation studies, the innovative module designs, and the model’s ability to maintain strong performance with high-speed inference. Reviewers xRnT and qxVi further noted that the paper is clearly written.


### Addressed Weaknesses and Questions

(1) *Reviewer XFjK: Concerns on multi-garment metrics, comparison with DreamVVT/SwiftTry, qualitative examples, and underperforming scenarios.*
We provided quantitative results for multi-garment scenarios, added comparisons with DreamVVT and SwiftTry, included more qualitative examples and video results, and discussed the underperforming scenarios. We believe that the rebuttal responses effectively addressed the reviewer’s concerns.

(2) *Reviewer HuPr: Architectural novelty, inference-time details, and complex ablation settings.*
We clarified that our goal is not to design an entirely new architecture but to address key challenges in video virtual try-on through several effective module designs. We added inference-time details and simplified ablations following the reviewer’s suggestion. We believe that the rebuttal responses effectively addressed the reviewer’s concerns.

(3) *Reviewer xRnT: Architectural choices, redundancy of semantic/structural/appearance tokens, and cross-category garment scenarios.*
We clarified the motivation behind our architectural choices and explained why the semantic/structural/appearance tokens are not redundant.
The revised version includes more cross-category garment scenarios and additional video examples.
We also extended discussions on the semantic/structural/appearance cue injection, the DMD component, and the mask-based pipeline.
We believe that the content provided in the rebuttal, along with the subsequent discussions, has resolved the reviewers’ questions and concerns.

(4) *Reviewer qxVi: Component choices, interpretability, ablation clarifications, and video results.* We provided explanations and additional experiments in the rebuttal to address each issue raised by the reviewer. We further discussed the comparison with Wan2.1-Fun-Control and the component choices with Reviewer qxVi, and resolved these concerns through additional qualitative and quantitative experiments.

### Revised Manuscript Updates
Regarding the revised version, we update the manuscript with the following additions:

1. Section 3.3 now provides a detailed description of the training objectives.
2. Section 4.3 adds a new subsection Comparison in Cross-Category Garment Scenarios, along with comparisons against DreamVVT and SwiftTry, and an additional comparison with Wan2.1-Fun-Control.
3. Section 4.4 includes two new subsections: Additive of Incremental Components and Line-Token Attention Map.
4. Appendix A.3 now includes Quantitative Metrics under Multi-Garment Scenarios.
5. Appendix A.13 adds the analysis of Line Estimation vs. Canny Edge.
6. Appendix A.14 adds the study on the Impact of Caption Quality.
7. Appendix A.15 includes additional discussions on Underperforming Scenarios.

We also provide extra video results in Supplementary_Material.zip, including Video Comparison with Other Methods.mp4, Ablation Study Video.mp4, and Cross-Category Garment Scenarios.mp4.

We summarize the main innovations of our MagicTryon as follows:

1. We decompose garment information into three attributes: semantic, structural, and appearance cues, and explicitly inject them into the denoising network. **This design is not a standard practice in existing methods such as ViViD and CatV2TON.**
2. We extend rotary position encoding to a garment-aware spatiotemporal RoPE, providing explicit cross-frame correspondence constraints that enable the model to preserve garment identity even under human motion.
3. We adopt distribution-matching distillation to compress the inference process to four diffusion steps while maintaining try-on quality, achieving a 50$\times$ speed-up. **The accelerated version of MagicTryOn not only improves practical usability but also enables our method to serve as an efficient baseline for future research.**
4. The experimental results validate the substantial performance improvements, **establishing MagicTryOn as an important baseline in the virtual try-on community and further advancing video virtual try-on**.

---

### Note · Authors · 2026-01-26

I have read and agree with the venue's withdrawal policy on behalf of myself and my co-authors.

---

### Meta-Review · Area_Chair_XXoj · 2026-01-07

**Summary:**

The paper proposes MagicTryOn, a diffusion transformer–based framework for garment-preserving video virtual try-on, with fine-grained garment cue decomposition, garment-aware spatiotemporal RoPE, and distilled fast inference.

Reviewers acknowledge the paper is clearly written, technically sound, and supported by extensive experiments.

However, the overall consensus is that the technical novelty is limited, as the method largely builds upon existing DiT-based VVT frameworks with incremental module design.

While empirical results are strong, they are not sufficient to outweigh concerns regarding architectural originality and system complexity. As a result, the paper does not meet the acceptance bar.

**Reviewer Concerns:**

Addressed by rebuttal:
- Missing comparisons with recent VVT methods (e.g., DreamVVT, SwiftTry).
- Lack of quantitative evaluation for multi-garment and cross-category scenarios.
- Clarification of inference-time efficiency and experimental protocol.
- Additional ablations and qualitative/video results.

Remaining concerns:
- Limited novelty: most reviewers view the approach as a careful integration of existing techniques rather than a fundamentally new framework.
- System complexity: reliance on many large pretrained components weakens the perceived elegance and accessibility of the method.
- Questions remain on whether the proposed token decomposition is the most minimal or principled design choice.

**Reviewer Scores:**

- Reviewer qxVi: Remains positive (Accept).
- Reviewer XFjK: Remains borderline reject.
- Reviewer HuPr: Core concerns on novelty persist; remains reject.
- Reviewer xRnT: Concerns largely addressed, but novelty reservations remain; stays borderline reject.

Overall assessment: Despite improvements after rebuttal, only one reviewer is clearly positive. The majority remain unconvinced on novelty grounds, leading to a final recommendation of Reject.

---

### Decision · Program_Chairs · 2026-01-26

Reject